# Transparent ultrahigh-molecular-weight polyethylene/MXene films with efficient UV-absorption for thermal management

Xianhu Liu [1,2,6], Wenrui Zhang[1,2,6], Xin Zhang[1], Zhengui Zhou[3], Chunfeng Wang [2,4], Yamin Pan [1] ✉, Bin Hu [3] ✉, Chuntai Liu [1] ✉, Caofeng Pan [2,5] ✉ & Changyu Shen[1]

The rational use and conversion of energy are the primary means for achieving the goal of carbon neutrality. MXenes can be used for photothermal conversion, but their opaque appearance limits wider applications. Herein, we successfully develop visible-light transparent and UV-absorbing polymer composite film by solution blending the MXene with polyethylene and then vacuum pressing. The resulting film could be quickly heated to 65 °C under 400 mW cm$^{-2}$ light irradiation and maintained over 85% visible-light transmittance as well as low haze (<12%). The findings of the indoor heat insulation test demonstrate that the temperature of the glass house model covered by this film was 6-7 °C lower than that of the uncovered model, revealing the potential of transparent film in energy-saving applications. In order to mimic the energy-saving condition of the building in various climates, a typical building model with this film as the outer layer of the window was created using the EnergyPlus building energy consumption software. According to predictions, they could reduce yearly refrigeration energy used by 31-61 MJ m$^{-2}$, and 3%-12% of the total energy used for refrigeration in such structures. This work imply that the film has wide potential for use as transparent devices in energy-related applications.

Could we live in a fossil-fuel-free world? This is one of the questions from "125 questions: Exploration and discovery", which was recently released by *Science*. The reduction in fossil energy reserves is an irreversible trend. According to the BP Statistical Review of World Energy[1], the remaining fossil fuels will last less than 100 years. Dependence on fossil energy is thus an unsustainable development path. To cope with this energy crisis, many kinds of energy sources have been discovered and developed. Solar energy is an inexhaustible and pollution-free energy source. The amount of solar energy reaching the Earth's surface per hour is greater than the amount of energy used by the relevant population over an entire year[2,3]. To make rational use of these alternative resources, people have begun to explore solar energy applications. Solar energy has been widely used in some fields, such as photothermal applications[4–6] and photoelectric conversion[7–9]. However, few studies have focused on ultraviolet light (UV). Ultraviolet light makes up only 7% of sunlight, yet it harms most life because of its

[1]College of Materials Science and Engineering, State Key Laboratory of Structural Analysis, Optimization and CAE Software for Industrial Equipment, National Engineering Research Center for Advanced Polymer Processing Technology, Zhengzhou University, Zhengzhou 450002, China. [2]Beijing Institute of Nanoenergy and Nanosystems, Chinese Academy of Sciences, Beijing 101400, PR China. [3]Wuhan National Laboratory for Optoelectronics, Huazhong University of Science and Technology, Wuhan 430074, PR China. [4]College of Physics and Optoelectronic Engineering, Shenzhen University, Shenzhen 518060, China. [5]Institute of Atomic Manufacturing, Beihang University, Beijing 100191, China. [6]These authors contributed equally: Xianhu Liu, Wenrui Zhang. ✉e-mail: yamin.pan@zzu.edu.cn; bin.hu@hust.edu.cn; ctliu@zzu.edu.cn; cfpan@binn.cas.cn

penetrating power. Given this problem, it is of great significance to develop materials that can convert ultraviolet light into usable energy through reasonable means[10]. Moreover, the development of transparent materials has become increasingly popular in recent years[11]; such development improves aesthetics and offers a broader choice of materials for items such as windows, optical devices[12], and transparent sensors[13–15]. However, high light transmittance means that most light will pass through such materials and cannot be efficiently absorbed. In contrast, solar devices are generally opaque (black or dark) to ensure adequate light absorption[16–18], but this monotonous color choice enormously restricts the application scope of such devices.

As an important kind of plastic, ultrahigh-molecular-weight polyethylene (UHMWPE) has been widely used in many fields. Additionally, it has been reported that ultra-drawn UHMWPE films with high molecular orientation can exhibit excellent transparency and good thermal conductivity[19]. Moreover, UHMWPE is one of the best choices for protective coatings under corrosion and abrasion conditions due to its low surface energy[20]. However, the ultrahigh-molecular weight of UHMWPE leads to difficulties in processing and filler dispersion, limiting its further application. Accordingly, it is still a challenge to prepare highly transparent UHMWPE composite films with desirable properties, and to our knowledge, there are few publications on such composite films available in the literature. To take advantage of the abovementioned benefits of UHMWPE, a method to facilitate processing to form UHMWPE composite films is being developed.

MXenes, distinguished by their versatile surface chemistry, feature functional terminations (-OH, -F, and -O) that are not typically present on other two-dimensional (2D) materials, having lots of ideal properties, including electrical conductivity[21,22], thermal conductivity[23,24], electromagnetic shielding properties[25–28], and the potential for photocatalysis applications[29–31]. Surprisingly, MXenes have nearly perfect photothermal conversion efficiency, as shown in previous studies[32–34]. The only downside is the opacity of MXenes, which mainly results from the multilayer structure that blocks light penetration. However, numerous reports on MXenes have confirmed through nuclear magnetic resonance spectroscopy that MXenes have many surface functional terminations[35–37]. These surface groups are pivotal for the material's optical properties, enabling the fine-tuning of their interaction with light. Specifically, the ability to modify these surface terminations allows for the precise control over the optical transparency of MXenes, a property of paramount importance for applications in optoelectronic devices, transparent conductive films, and energy-efficient windows[38,39].

Herein, a transparent composite film with photothermal conversion performance was developed by mixing MXene@2-(2H-Benzotriazol-2-yl)−4,6-ditertpentylphenol (BZT) with UHMWPE. The combination of the MXene and BZT was shown to improve the transparency of the composite film because BZT promoted the dispersion of MXene in the matrix, and BZT exhibited excellent compatibility with UHMWPE. Thus, this composite film, which exhibited both excellent photothermal properties and low haze, has immense potential for thermal management in transparent device applications.

## Results
### Fabrication strategy and characterization of transparent composite films
To prepare a transparent material with excellent photothermal properties, the MXene component by etching $Ti_3C_2T_x$ was prepared. The preparation mainly involves etching the Al layer in $Ti_3AlC_2$ and then centrifuging the material to obtain a few layers of MXene. 1D-WAXD spectra of $Ti_3AlC_2$ and $Ti_3C_2T_x$ MXene are shown in Supplementary Fig. 1. The peak angle of $Ti_3AlC_2$ (002) was observed to shift from 9.72° to 6.12° upon etching, indicating that the layer spacing of $Ti_3AlC_2$ had increased; according to the Bragg formula ($2dsin\theta = n\lambda$, where $d$ represents the layer spacing, $\theta$ is the Bragg angle, and $n$ is the

diffraction order; in this experiment, the X-ray wavelength $\lambda$ is 0.154 nm), the layer spacing before and after etching was calculated to be 9.12 Å and 14.45 Å, respectively. This significant increase in layer spacing indicates the successful etching of $Ti_3C_2T_x$ MXene. The SEM image shown in Supplementary Fig. 2a reveals a conventional 2D structure of MXene nanosheets with a size of 1–2 μm, and this result was further confirmed by the AFM image shown in Supplementary Fig. 2b. The thickness distribution along the direction indicated by the white dashed lines shows that the thickness of the MXene sheet was between 2 and 3 nm (the average thickness of single-layer MXene is approximately 0.98 nm[40,41]). Meanwhile, the XPS diagram (Supplementary Fig. 3) shows that the $TiO_2$ and $TiO_2$-F groups in MXene account for about 10% of the total O groups. Thus, the above results demonstrate that MXene nanosheets with a few layers were prepared successfully.

Next, UHMWPE films containing different amounts of MXene and BZT were prepared by solution blending and hot pressing (Fig. 1a) to ensure the complete combination of the MXene and BZT. The prepared composite films were labeled xMyB, where x and y represent the percentage contents of MXene and BZT, respectively. The crystallinity and crystallization temperature after compression is much lower than those of UHMWPE film before compression (Fig. 1b, c), which is due to the re-melting and rapid cooling of the film during the vacuum compression molding process, making the molecular chain segment movement blocked and crystallization incomplete. Similarly, the crystallinity of the composite film with filler is lower than that of the pure UHMWPE film, because the molecular chains are entangled in the MXene lamellar structure, and the addition of too much filler will restrict the movement towards the chains, which is not conducive to the formation of crystals[42]. Next, to explore the influence of the addition of MXene and BZT on the thermal stability of the composite films, TGA and DTG curves were obtained and are shown in Fig. 1d, e, respectively. Here, the temperature corresponding to a residual mass of 90% was defined as the thermal decomposition temperature ($T_d$). The 0.5M0B film (449.69 °C), 0M2B film (450.43 °C), and 0.5M2B film (453.10 °C) demonstrated considerable improvements in $T_d$ compared to pure UHMWPE (424.05 °C), demonstrating that MXene and BZT increased the films' overall thermal stability. However, $T_{MAX}$ (peak of the DTA curve) of 0.5M0B film (493.41 °C) is higher than that of 0.5M2B film (486.30 °C), because the undispersed MXene with large flake sizes tend to make the blocking effect more effective.

To evaluate the mechanical properties of the prepared composite films, stress-strain experiments were performed. The results obtained before and after the films were pressed are shown in Fig. 1f, g, indicating that the pressing process significantly improved the mechanical strength of the films. This improvement was mainly attributed to a large number of pores inside the films, which were extraordinarily reduced after pressure was applied, making the inner structure of the UHMWPE film more compact. An evaluation of the UHMWPE films with different MXene loadings revealed that the tensile strength (σ), elongation at break (ε), and tensile modulus (E) of the composite films were all significantly improved relative to those of the pure UHMWPE film. For example, the σ, ε, and E values of the 0.1M2B film were 22 MPa, 554%, and 129.89 MPa, respectively, which are much higher than those of the pure UHMWPE film (14.2 MPa, 218.5%, and 74.3 MPa, respectively). This is due to the fact that when strain is applied, the tension in the UHMWPE matrix is transferred to MXene, whose unique two-dimensional laminate structure is able to withstand the strain and absorb the energy from external pressure forming strong interfacial interaction with UHMWPE[10,43], thus preventing the creation and expansion of cracks. However, with a further increase in the MXene content, the mechanical properties of the composite films decreased gradually. This is because excessive MXene introduced more defects, negatively impacting the overall dispersion and thus reducing the mechanical performance.

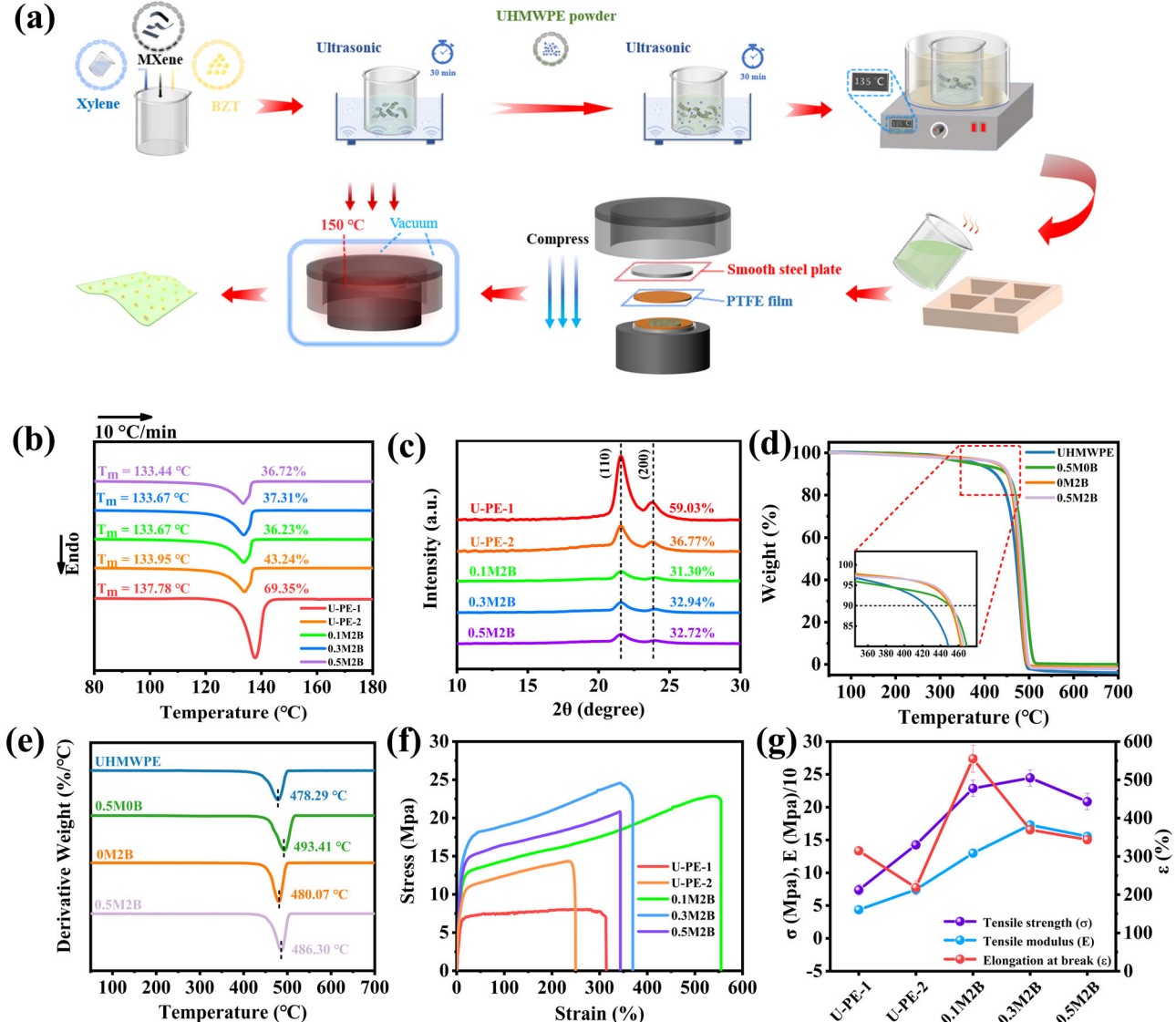

**Fig. 1 | Fabrication process, characteristics and mechanical properties of the UHMWPE composite films. a** Schematic diagram of the preparation process of MXene@BZT/UHMWPE films. **b** DSC and **c** 1D-WAXD curves of composite films with 2 wt.% BZT and different MXene contents. **d** TGA and **e** DTG curves of composite films with different contents. **f** Stress-strain curves and **g** corresponding calculation results for composite films before and after application of pressure, where U-PE-1 and U-PE-2 represent UHMWPE films before and after being pressed, respectively. The error bars represent the standard deviations of the measured values.

## Dispersing mechanism of transparent composite films

The optical images of 0.5M and 0.5M2B are shown in Fig. 2a. After the addition of BZT, the composite film filler particles significantly decreased in size and were more uniformly distributed without obvious agglomeration (Fig. 2b). Dispersions of MXene alone and MXene mixed with BZT in xylene are shown in Supplementary Fig. 4a–f. A precipitate formed in the dispersion solution of MXene alone after 1800 s, while the mixture of MXene and BZT together was still well dispersed after 3600 s. In addition, the UHMWPE composite film formed by adding MXene alone exhibited obvious agglomeration (Supplementary Fig. 4g), whereas uniform dispersion in the composite film was observed after the addition of BZT. The combination of the two is shown in Fig. 2c. MXene has a two-dimensional lamellar structure and a large specific surface area, whereas BZT, as a needle-like crystal, can be attached to the surface of MXene lamellae by bonding. BZT is compatible with UHMWPE, thus improving the dispersion of MXene in the matrix. The hybridization mode of MXene@BZT was determined by FTIR (Fig. 2d). Compared with the films containing BZT and MXene alone, the spectra of MXene@BZT/UHMWPE films showed a significant red shift at 3350 cm$^{-1}$ and 1594 cm$^{-1}$, corresponding to the absorption peaks of -OH and Ti-O, triazole, respectively. In addition, the same general trend was observed in the comparison of the infrared spectra of MXene, BZT powder, and their mixtures (Supplementary Fig. 5). These results suggest that MXene and BZT form the interaction of hydrogen bond.

## Optical properties of transparent composite films

Next, the optical properties of the composite films were measured by an ultraviolet-visible spectrophotometer, as shown in Fig. 3a, b. The light absorption and transmittance were affected by the film thickness, as shown in Supplementary Table 1. The transmittance of the composite films before and after being pressed was compared. The results show that the transparency of the composite films after pressing improved to a certain extent (approximately 10% to 20%), which was attributed to the decrease in light scattering due to the reduction in pores in the film created by evaporation of solvent. Moreover, the

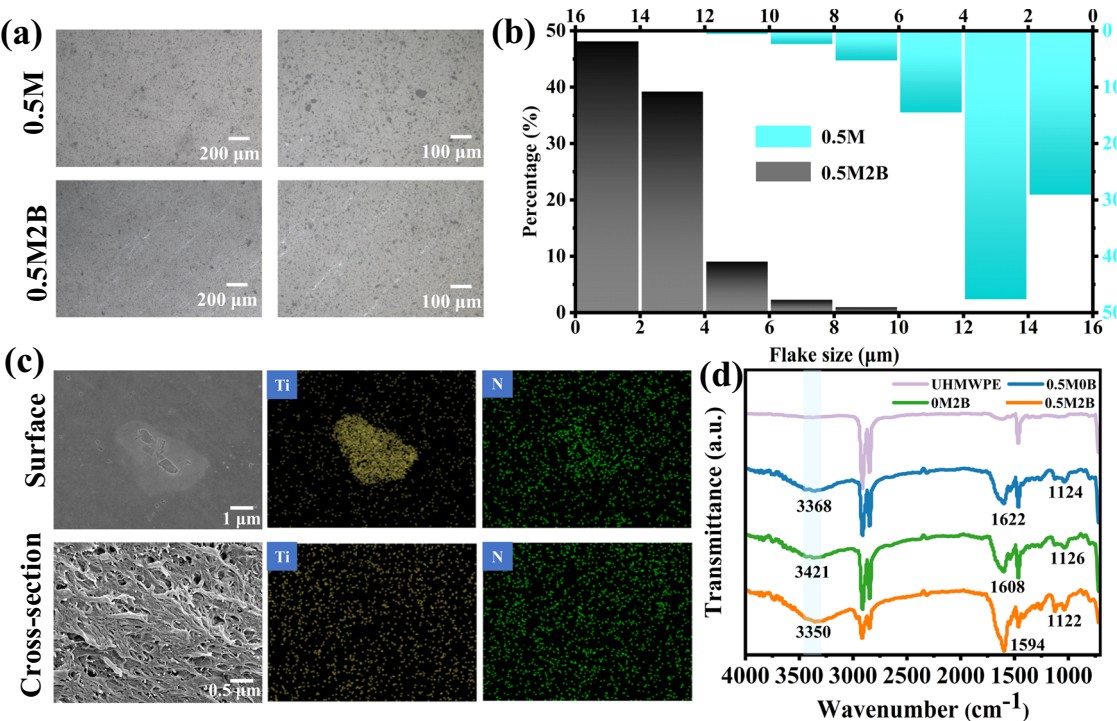

**Fig. 2 | Dispersity and dispersing mechanism of the UHMWPE composite films. a** Optical images and **b** flake size distributions of composite films. **c** SEM and EDS of surface and cross-section of 0.5M2B film. **d** FTIR spectra of composite films with different contents.

addition of the MXene filler had a negative effect on the transparency of the films due to the light impenetrability of the MXene. For this reason, the effects of different BZT loadings on the film transparency were further studied, and the transmittance curves are shown in Supplementary Fig. 6. Adding 2 wt.% BZT to 0.3 wt.% MXene significantly improved the transparency of the film, and the transparency further increased as the BZT content increased from 2 wt.% to 8 wt.%. The UV shielding performance of the films was explored by comparing the transmission spectra of Supplementary Fig. 7 for the composite films with MXene and BZT separately. As a material with excellent photothermal conversion efficiency, MXene is well absorbed in the solar energy band. At the same time, the band gap of $TiO_2$ produced by MXene oxidation is between 3.2 and 4.0 eV, which enables it to match photon energy well and produce effective UV shielding[44]. The triazole and the hindered phenol functional group of BZT molecules provide a strong shielding effect of UV light at 306 and 347 nm. These results suggest the good application potential of UHMWPE composite films filled with MXene and BZT in the field of anti-UV transparent devices.

Another essential indicator of optical performance is haze. One characteristic of UHMWPE films is high haze, which makes them appear very different from glass even though both materials have high light transmittance. The haze of the composite films was measured by the four-step method (Supplementary Table 2), and the results are shown in Fig. 3c. Significant changes occurred before and after the films were pressed, and the haze values decreased from more than 95% to a minimum of 12%, which is rare for UHMWPE and its composites. It can be inferred from the SEM images (Fig. 3d) that the main reason for the low haze of the films may be the decrease in surface roughness. The surface roughness of the films prepared in this study is largely due to the evaporation of the solvent, leaving a large number of voids, while the surface roughness of the films obtained after melting and pressing was enormously reduced since the melt filled the voids and was pressed tightly. Moreover, the addition of BZT effectively reduced the haze of the pressed films. Among the considered samples, 0.3M4B and 0.1M2B exhibited the lowest haze values and probably represent the

optimal ratio range in which BZT promotes MXene dispersion. In numerous reports, films have been shown to be transparent by attaching them to a background image, but this method does not indicate transparency entirely accurately. Therefore, the composite films in this study were placed in two configurations (Fig. 3d). In the first case, the films were placed directly on the image, and in the second case, the films were placed at a certain height above the image. Two films with vastly different haze values both showed high transparency in the first case, but only the films with low haze also showed transparency in the second case. These results also suggest that these composite films have improved potential for applications such as transparent glass coatings.

In Fig. 3e, the transmittance and haze of the composite film developed in this study are compared with those of other polyethylene-related films, and the details are shown in Supplementary Table 3. The haze of previously reported UHMWPE films is more than 50%, and the transmittance is rarely more than 80%. High transmissivity and low haze are commonly observed in linear low-density polyethylene (LLDPE) composite materials, but haze values of more than 20% still occur, mainly caused by differences in molecular weight and crystallinity. This relationship is attributed to the regular arrangement of molecules and the lower crystallinity, which help reduce the scattering of light in the film and improve the transmittance. The UHMWPE-based composite films developed in this research have high light transmittance and low haze, and these properties are better than those of most UHMWPE products and even LLDPE[45–49]. Compared with the latter, UHMWPE exhibited improved mechanical properties and corrosion resistance, so UHMWPE products with excellent optical properties are highly desirable.

## Photothermal conversion of transparent composite films
After the transparency characterization, the photothermal conversion performance of the composite films was investigated. Pure UHMWPE and its composite films were exposed to simulated sunlight. To ensure effective photothermal conversion, the films were exposed to light

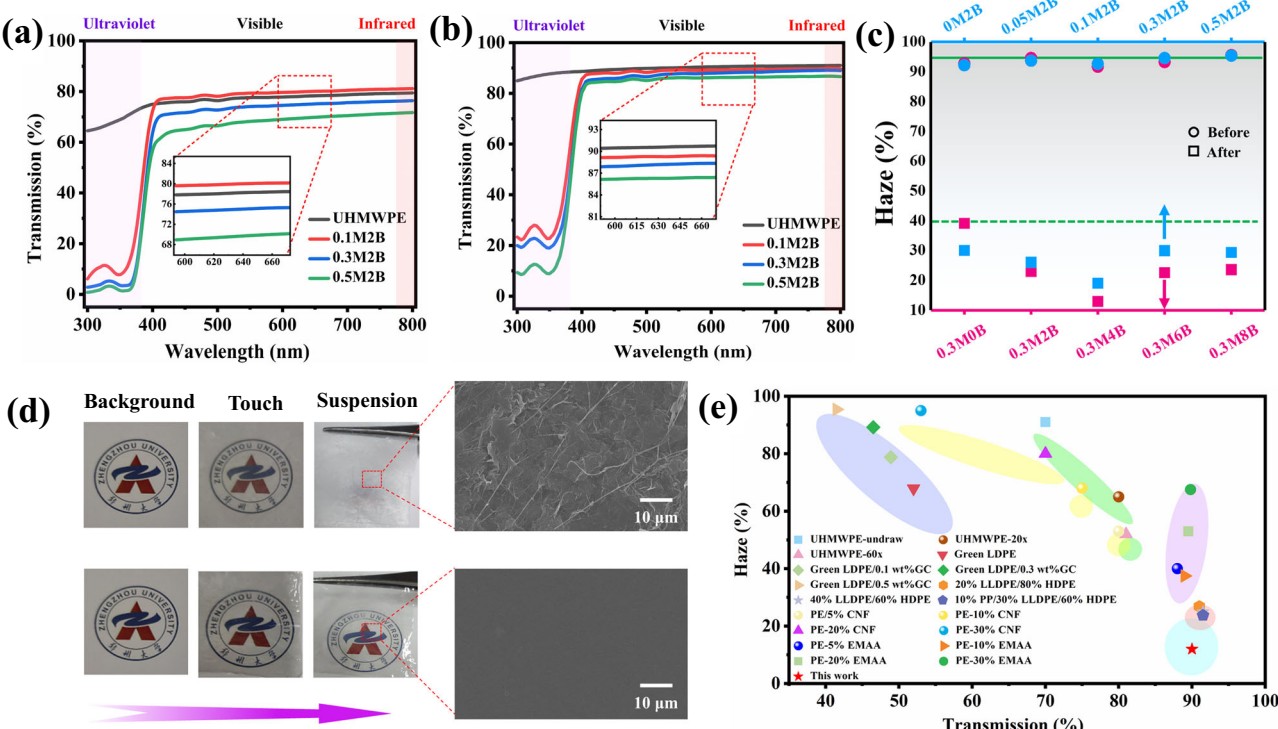

**Fig. 3 | Optical properties of the UHMWPE composite films. a, b** UV-vis transmission spectra of composite films with different MXene contents before (**a**) and after (**b**) being pressed. **c** Haze of composite films with different contents of MXene and BZT before and after being pressed. The solid and dashed lines correspond to UHMWPE films before and after being pressed, respectively. **d** Optical photographs and SEM images of composite films before (top) and after (bottom) being pressed when placed in different configurations. **e** Overview of haze and transmittance of polyethylene products. Data for comparison have been obtained from the literature.

with power densities of 400 mW cm$^{-2}$ (strong) and 100 mW cm$^{-2}$ (weak) for 80 s. The changes in surface temperature with irradiation time are shown in Fig. 4a, b. The temperature of the films containing MXene increased significantly after light irradiation, while the temperature changes of the pure UHMWPE film were minor (an increase of less than 2 °C in 80 s at 100 mW cm$^{-2}$ and only approximately 6 °C at 400 mW cm$^{-2}$). Specifically, the temperature of the 0.5M2B film increased by 6 times under the same conditions; the detailed IR images are shown in Supplementary Figs. 8 and 9. This large difference was attributed to the good dispersion and excellent photothermal conversion performance of MXene in the UHMWPE matrix, which maximizes the ability of the film to convert the incident light into heat energy.

To further explain the origin of this photothermal conversion capability, UHMWPE films containing 0.3 wt.% MXene and different BZT contents were exposed to light under the same conditions described in the previous paragraph. The curves of the temperature change over time on the surface of the films are shown in Fig. 4c. Surprisingly, the composite films also showed photothermal conversion upon irradiation with light of different power densities, and the film surfaces exhibited a certain range of temperature increases with increasing BZT content. However, the photothermal conversion ability of 0.3M8B composite film is lower than that of 0.3M6B composite film. This trend was also observed at 100 mW cm$^{-2}$ (Supplementary Fig. 10). The promotion of photothermal conversion by BZT can be explained as follows: On the one hand, as a strong ultraviolet absorber, BZT may compete with MXene's absorption in the ultraviolet band; on the other hand, as a dispersant of MXene, BZT can reduce its agglomeration and enhance the photothermal conversion efficiency of MXene. To further verify the experimental results, the photothermal capacities of the composite films with MXene alone and BZT alone were measured (Supplementary Fig. 11). The results showed that the heating effect of

BZT was negligible and that the photothermal conversion ability mainly depended on the MXene. A good linear relationship between light intensity and equilibrium temperature was observed, as shown in Fig. 4d. This finding was attributed to the excellent dispersion and stability of MXene and BZT in the UHMWPE matrix and the protective effect of the UHMWPE film. In summary, this property demonstrates the controllability of the MXene@BZT/UHMWPE film's photothermal behavior.

To evaluate the stability of the photothermal conversion of the composite films, five light on/off cycles under different light power densities were performed, with each cycle comprising an irradiation duration of 20 s and then a period of 20 s with the light switched off, as shown in Fig. 4e, f. The temperature improvement efficiency of the films did not deteriorate with increasing cycle number, which indicates that the photothermal conversion of these composite films can occur in cyclic processes without fatigue.

### UV-absorption of transparent composite films

The high transparency and photothermal conversion of MXene@BZT/ UHMWPE films are mainly attributed to the selective absorption of light (Fig. 5a). Light with a wavelength greater than 400 nm (mainly composed of visible light) is rarely absorbed by such films, which is the reason for their excellent transparency. The light that is effectively absorbed and converted into heat by these films is mainly ultraviolet light with wavelengths less than 400 nm. The absorbance spectra of the UHMWPE films further confirmed this inference (Fig. 5b, c). When the wavelength applied to the UHMWPE composite films was less than 400 nm, the light absorption of the films sharply improved. This finding suggests that MXene@BZT/UHMWPE films can maximize photothermal conversion, especially in locations with severe ultraviolet radiation hazards. To examine the feasibility of practical applications of the composite film, the composite film was placed on a wrist

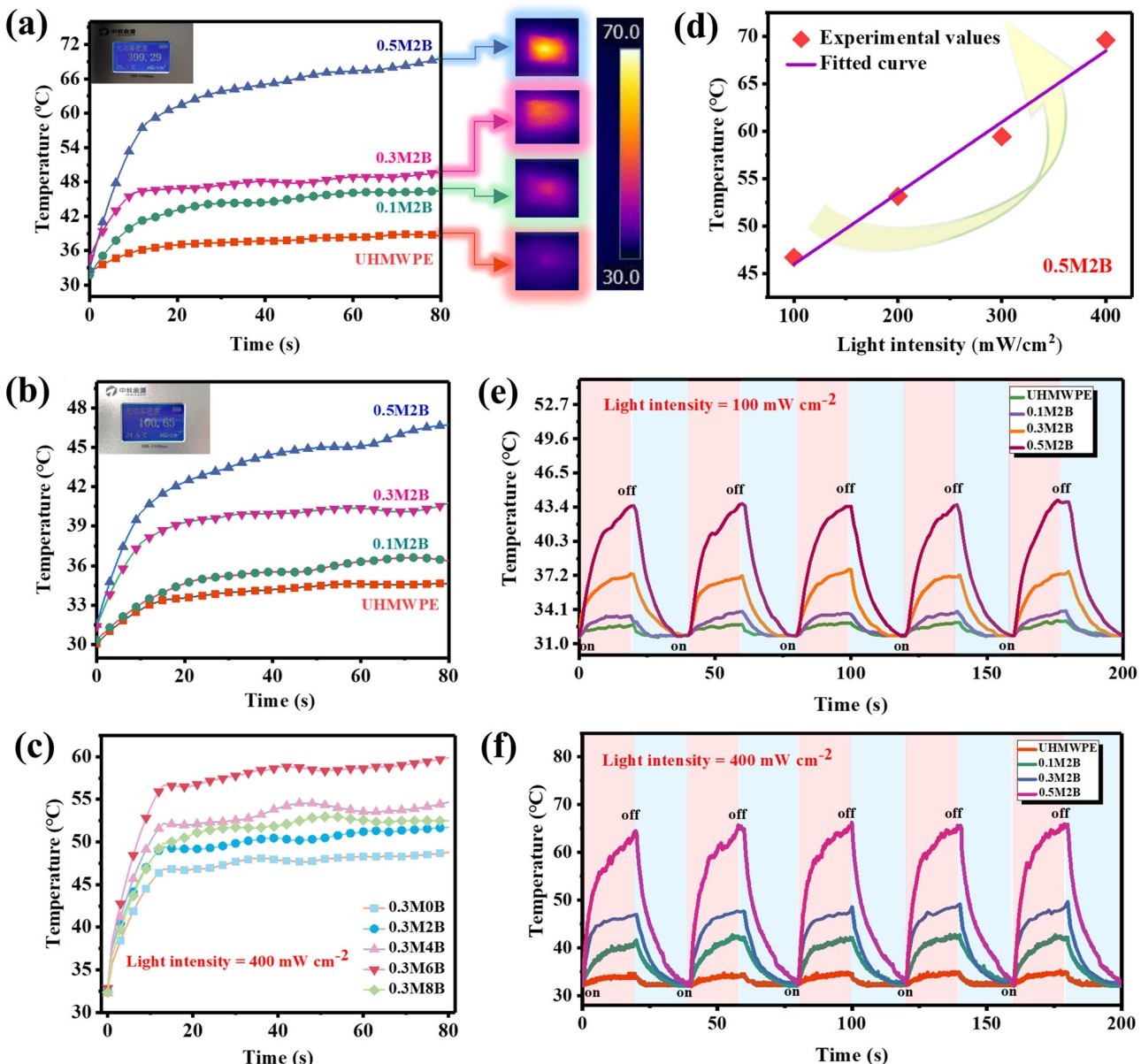

**Fig. 4 | Photothermal conversion of the UHMWPE composite films.**
**a**, **b** Temperature-time curves of UHMWPE films with different MXene contents under light irradiation of 400 mW cm⁻² (**a**) and 100 mW cm⁻² (**b**). The illustration on the right side of (**a**) shows the corresponding IR thermal images. **c** Temperature-time curve of UHMWPE films with different BZT contents subjected to 400 mW cm⁻² irradiation. **d** Experimental data and results of linear fitting for the equilibrium temperature under different light intensities. **e**, **f** Cyclic temperature rise curves of UHMWPE films with different MXene contents subjected to 100 mW cm⁻² (**e**) and 400 mW cm⁻² (**f**) irradiation.

and irradiated for 60 s at 100 mW cm⁻². IR images of the wrist and film surface temperatures are shown in Fig. 5d. One minute after exposure, the temperature of the composite film reached 40 °C, and the wrist temperature was only 31.5 °C, indicating that the composite film still has excellent photothermal transformation ability in practical applications.

## Cooling performance of transparent composite films and their modeling cooling energy savings

To test the temperature control ability of the UHMWPE composite films, three identical model devices consisting of foam, glass containers, and sensors were placed in a real outdoor environment (Fig. 6a, b). The model devices were covered with polypropylene foam to reduce the influence of thermal convection and radiation from outside. The temperature, humidity, and illumination inside the

devices covered with 0.5M2B films (4 × 4 cm² (partial coverage) and 8 × 8 cm² (full coverage)) or no film (blank) were measured (Fig. 6c). The addition of the 0.5M2B film effectively inhibited the temperature increase observed in the blank group. When the composite film completely covered the glass, the temperature inside the container was reduced by 6–7 °C, and this reduction increased with increasing illumination. As shown in Fig. 6d, the application of the composite film with complete coverage of the device effectively reduced the temperature inside, suggesting that these composite films are more suitable for large-window buildings such as skyscrapers. Another important consideration is whether the application of the film to a window will affect the illumination of the interior space. As shown in Supplementary Fig. 12, compared with that of the blank device, the illumination of the 0.5M2B film remained above 80% and did not decrease with the extension of the test time. These results show that

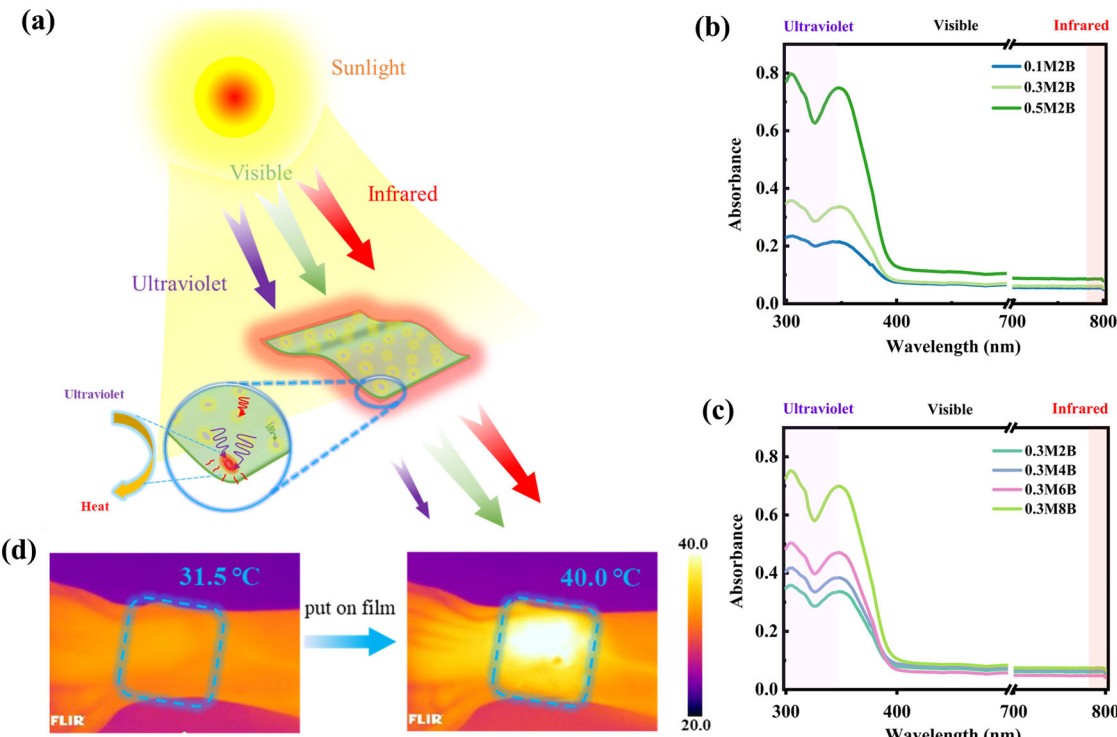

**Fig. 5 | UV-absorption of the UHMWPE composite films. a** Schematic diagram of photothermal conversion by the UHMWPE composite film. **b, c** Absorbance spectra of UHMWPE films with 2 wt.% BZT and different MXene contents (bottom) and UHMWPE films with 0.3 wt.% MXene and different BZT contents (top). **d** IR thermal images of a wrist and a 0.5M2B film attached to the wrist after irradiation for 1 min at 100 mW cm⁻².

the composite film can be applied to windows without affecting indoor passive lighting. This outdoor test experimentally demonstrates that it is possible to achieve radiative cooling by using the UHMWPE composite film, and such films are expected to be more widely used in the future due to their high transparency.

To evaluate the practical application value of the composite films, a typical midrise building model was used to simulate the cooling energy-saving potential. The annual cooling energy consumption in Phoenix, AZ, USA, which has a tropical desert climate, is shown in Fig. 7a. The cooling energy saved by using the UHMWPE film compared with bare glass was only 18.8 MJ m⁻², which is mainly due to the low selectivity of the UHMWPE film for light transmission. The composite films strongly absorb ultraviolet light; the cooling energy saved by the 0.5M2B film was as high as 61.3 MJ m⁻², which is 3.3 times that of the UHMWPE film, and the cooling energy-saving percentage was increased from 1.3% to 4.2%. Due to the high intensity of ultraviolet light in summer, the monthly cooling energy savings are higher in summer than in winter (Fig. 7b). These results indicate that composite film-covered windows have excellent cooling energy-saving potential in hot climates.

The cooling energy savings of the building model using data from 12 cities around the world (Fig. 7c) were evaluated. The cities are located on five continents with nine climatic conditions based on the Köppen climate classification (Supplementary Table 4). The simulated results show that the cooling energy savings of the 0.5M2B film are between 31.2 and 61.3 MJ m⁻² (Supplementary Fig. 13), considering Beijing, China (31.2 MJ m⁻²); Turpan, China (40.6 MJ m⁻²); Wuhan, China (22.2 MJ m⁻²); New Delhi, India (56.5 MJ m⁻²); Singapore (56.2 MJ m⁻²); Phoenix, AZ, USA (61.3 MJ m⁻²); Atlanta, GA, USA (46.6 MJ m⁻²); Athens, Greece (45.5 MJ m⁻²); Rome, Italy (38.2 MJ m⁻²); Canberra, Australia (43.6 MJ m⁻²); Mexico City, Mexico (47.8 MJ m⁻²); and Johannesburg, South Africa (51.9 MJ m⁻²). The simulated results show that the UHMWPE composite film has an excellent cooling energy-saving effect compared with the pure UHMWPE film and that the energy-saving effect is significantly enhanced as the content of MXene increases

(Supplementary Figs. 14–16). In addition, the UHMWPE composite film plays a greater role in cooling energy savings in hotter climates than in cooler climates. However, the energy-saving percentages are low in cities in tropical zones, such as Singapore (2.9%), New Delhi (3.7%), and Phoenix (4.2%). The highest energy-saving percentage was observed for Canberra, which has a temperate oceanic climate with cool summers. In these cities, the cooling energy consumption is relatively low, and thus, the role of the composite film in energy savings is more prominent than that in other cities. However, it is worth noting that a reduction in cooling power consumption means that heating consumption will increase in cold winter conditions. In contrast to other coatings, the composite film developed in this study can be reasonably removed or rolled up to overcome this issue.

Finally, the anti-aging performance of the composite film was tested (based on ASTMF 1980:2002). The aging temperature selected for the accelerated aging test was 60 °C, the preset aging time was 72 h, and the ultraviolet spectrum irradiance (UVA-340) was 0.72 W/m². As shown in Supplementary Fig. 17, obvious cracks appeared in the UHMWPE film, while no obvious defects appeared in the 0.5M2B film. The visible light transmittance of the composite film was basically unchanged before and after the test, while the UV shielding performance was reduced (Supplementary Fig. 18). After accelerated aging, the mechanical properties of the UHMWPE films were very fragile and cracked, while the mechanical properties of the 0.5M2B films decreased but still had excellent aging resistance compared to the pure UHMWPE films (Supplementary Fig. 19).

## Discussion

A method is proposed to construct a MXene-filled UHMWPE film with high transparency and low haze for use in pollution-free photothermal conversion and energy savings. The hydrogen bond interaction of MXene and BZT increased the MXene's dispersion in the UHMWPE matrix, as well as the addition of MXene and BZT to create a two-dimensional laminate structure network considerably improved the

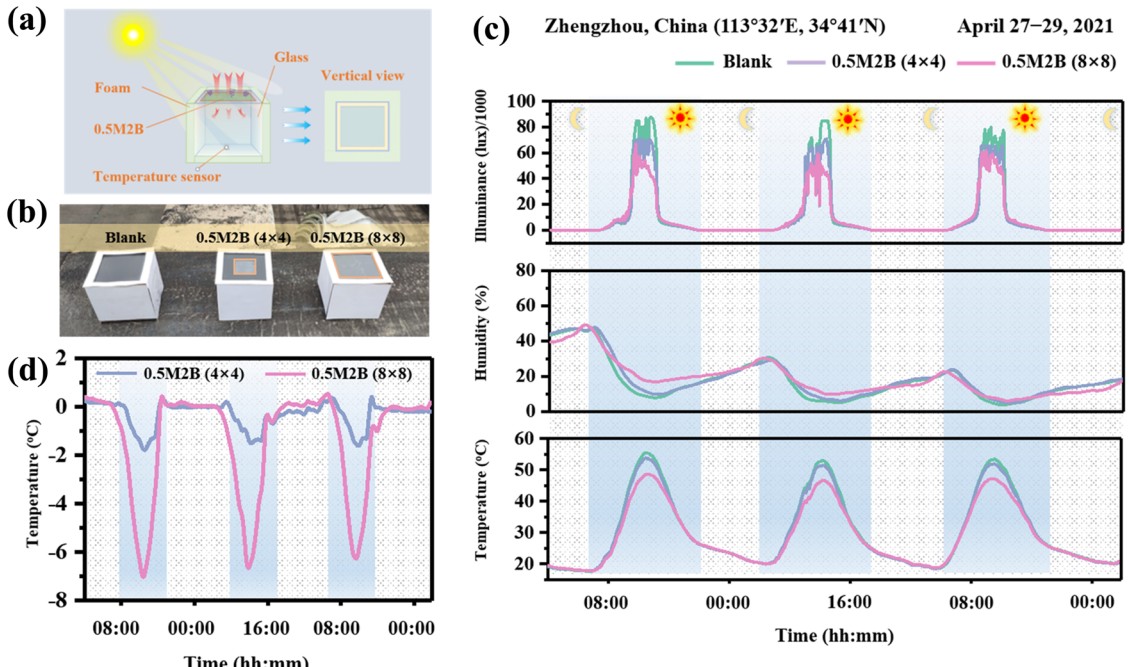

**Fig. 6 | Cooling performance of the UHMWPE composite films in comparison with bare glass. a, b** Diagram (**a**) and digital photograph (**b**) of the outdoor cooling performance measurement system. **c** Measured outdoor solar illumination (top), relative humidity (middle), and temperature (bottom) in Zhengzhou, China. **d** Cooling properties of 0.5M2B films with different sizes.

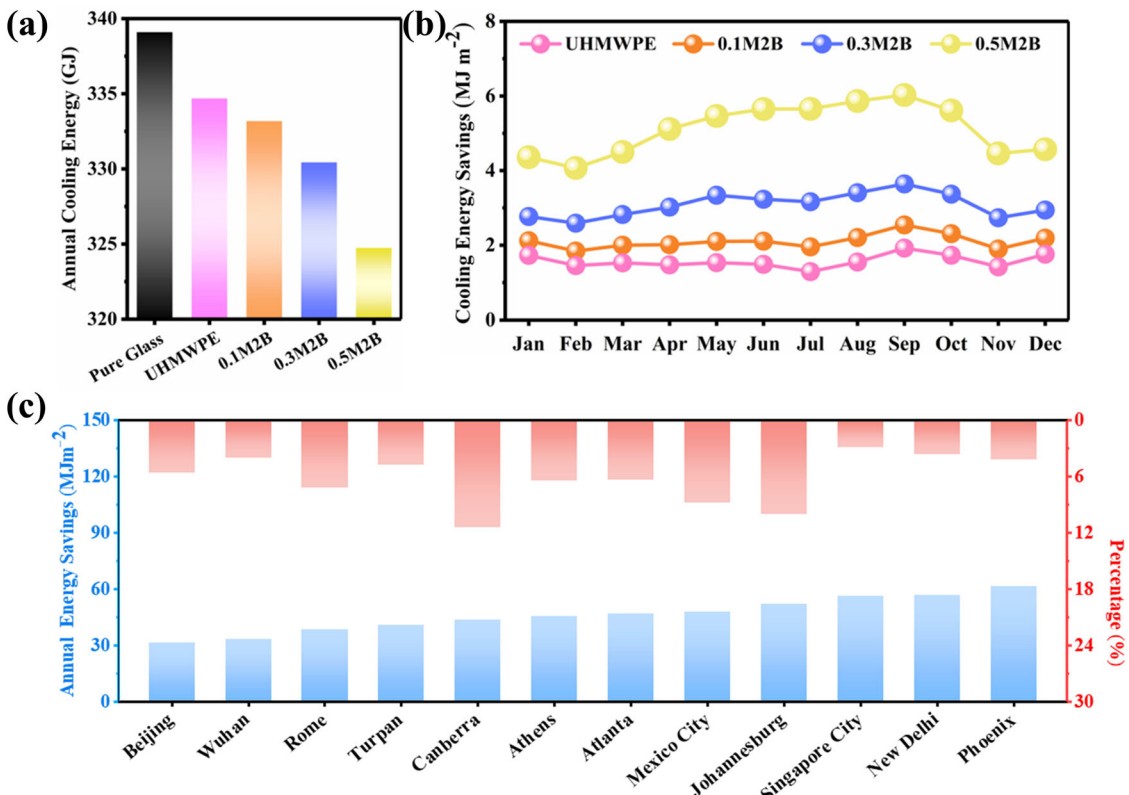

**Fig. 7 | Modeling cooling energy savings through window covering. a** Annual cooling energy consumption in building models using four types of windows based on weather data for Phoenix, AZ, USA (33.45°N, 111.98°W). **b** Monthly extra cooling energy savings of building models using these four types of windows. **c** Annual cooling energy savings and percentage for building models using the 0.5M2B film based on weather data from 12 cities.

mechanical characteristics and thermal stability of films. Moreover, as the matrix, UHMWPE has a remarkable protective effect on the internal filler. Because of the combination of these properties, under mild sunlight, a film with only a small amount of filler can absorb 78% of UV light while maintaining high transparency and stability. These films could be quickly heated to 65 °C under light irradiation of 400 mW cm$^{-2}$ and maintained over 85% visible light transmittance as well as low haze (<12%). The findings of the indoor heat insulation test demonstrated that the temperature of the glass house model covered by composite films was 6–7 °C lower than that of the uncovered glass house model. This film concept has broad application potential in the field of window and display screen coatings as well as thermal management.

The simple streamlined preparation process also demonstrates the technical feasibility of scalable production. In addition, because of the various types of polyethylene, UHMWPE can be easily replaced by low molecular weight polyethylene or other polyolefin, such as polypropylene. Therefore, with appropriate polymer processing technologies especially for film casting, composite films with well-dispersed MXene could be widespread manufacture and used. Accordingly, we expect that the MXene composite films hold promise for future energy-efficient and sustainable building applications, enabling a substantial reduction in carbon emission and energy consumption.

## Methods

### Materials
UHMWPE powder with an $M_n$ of $(2.0–3.0) \times 10^6$ g mol$^{-1}$ was supplied by Beijing Eastern Petrochemical Co., Ltd. Ti$_3$AlC$_2$ MAX (400 mesh, >99% purity) was purchased from Jilin 11 Technology Co., Ltd. The antioxidant Irganox 1010 was purchased from Dinghai Plastic Chemical Co., Ltd. BZT was purchased from Shanghai Aladdin Bio-Chem Technology Co., Ltd., China. Xylene (analytical reagent, 99%) was purchased from Damao Chemical Reagent Factory (Tianjin, China). All reagents were used directly without further purification.

### Synthesis of Ti$_3$C$_2$T$_x$ MXene sheets
Ti$_3$C$_2$T$_x$ MXene sheets were prepared by etching the aluminum layer of Ti$_3$AlC$_2$ MAX with reference to a previous work[50]. In summary, 2 g of LiF was added to 40 ml of hydrochloric acid (9 M) in a PTFE container and stirred at 400 rpm for 30 min. Then, 2 g of Ti$_3$AlC$_2$ MAX powder was slowly added to the container and allowed to react for 24 h at 35 °C. The etched MXene was washed with deionized water, followed by ultrasonication (150 W, 10 min) and centrifugation (2191 × g, 10 min) several times until the pH value was higher than 5. An appropriate amount of deionized water was added to the dispersion solution, followed by ultrasonication for 30 min and centrifugation for 10 min. The dark-brown upper liquid was collected and freeze-dried for 72 h to acquire MXene sheets.

### Preparation of MXene@BZT/UHMWPE films
The composite films were prepared by solution blending and hot pressing. The desired contents of MXene and BZT were ultrasonicated in 80 ml of xylene for 30 min. Then, 300 mg of UHMWPE powder and Irganox 1010 (0.05 wt.% relative to UHMWPE) were added to the dispersion and ultrasonicated for 30 min, and the mixture was stirred in an oil bath at 135 °C for 2 h until the solution was colorless and transparent. Then, the solution was poured into a PTFE mold, and the solvent was evaporated at room temperature to procure the film. Finally, the obtained film was heated for 10 min in a vacuum press (150 °C, 39.8 kN) under constant pressure.

### Characterization
Attenuated total reflectance Fourier transform infrared (ATR-FTIR) spectroscopy was performed by means of a Thermo Fisher Nicolet 6700 (USA) instrument in the range of 4000–400 cm$^{-1}$. 2D wide-angle X-ray diffraction (WAXD) was performed on a Bruker D8 Discover diffractometer at a scanning rate of 6° min$^{-1}$ over a scanning range of 2–31°. Atomic force microscopy (AFM) images were captured using a Bruker MultiMode 8. Differential scanning calorimetry (DSC) on a TA Instrument Q2000 calorimeter was used to study the thermal and crystallization properties of the composite films. The films were heated or cooled at a rate of 10 °C/min within the temperature range of 40–180 °C. Thermogravimetric analysis (TGA) and derivative thermogravimetry (DTG) were performed on a Perkin Elmer Pyris 1 (USA). The resulting curves were used to determine the thermal stability of the films in the temperature range of 30–700 °C under a stable temperature gradient (10 °C/min). The whole measurement process was performed under a nitrogen atmosphere. Stress-strain tests were conducted on a Shimadzu tensile testing machine (Japan) at a rate of 10 mm/min to analyze the mechanical properties of the composite films. Transmittance spectra were obtained on a UV-vis NIR absorption spectrometer (Cary 5000) over a wavelength range of 300-800 nm with an interval of 1 nm. The photothermal conversion was characterized by placing the films in a sunlight simulator (CEL-HXF300) and using an IR thermal imaging instrument (FLIR, E60) to capture images in real time. The light intensity was determined by means of an optical power meter (CEL-FZ-A).

### Theoretical simulation
EnergyPlus (version 9.1.0) software was used to simulate the energy consumption to evaluate the energy-saving effect of the MXene@BZT/UHMWPE films. The building model used in the simulation was a typical midrise apartment building[51], and the calculation parameters, such as the cooling setpoint (23.9 °C), heating setpoint (21.1 °C), coefficient of lighting consumption and internal load, were modified based on U.S. Department of Energy commercial reference building models. The total building area was 2350.96 m$^2$, the net conditioned building area was 2117.98 m$^2$, and the window opening area was 234.08 m$^2$. Online weather data were used to calculate the annual energy consumption of the building model using data for 12 cities around the world[52]. The cities are located on five continents and experience nine climatic conditions based on the Köppen climate classification. The effect of thermal resistance was neglected due to the negligible thickness of the composite film, and the measured UV-vis-NIR and IR spectra were used in the simulation.

The $T_{solar}$ (0.3–2.5 μm) and thermal emittance ε (2.5–20 μm) were calculated by using Eq. (1)[53] and (2)[54], respectively:

$$T_{solar} = \frac{\int_{0.3}^{2.5} d\lambda \cdot t(\lambda) \cdot I_{AM1.5}(\lambda)}{\int_{0.3}^{2.5} d\lambda \cdot I_{AM1.5}(\lambda)} \tag{1}$$

$$\varepsilon = \frac{\int_{2.5}^{20} d\lambda \cdot \varepsilon(\lambda) \cdot I_{BB}(T,\lambda)}{\int_{2.5}^{20} d\lambda \cdot I_{BB}(T,\lambda)} \tag{2}$$

and visible light transmittance $T_{vis}$ (0.4–0.76 μm) was calculated by means of Eq. (3)[55]:

$$T_{vis} = \frac{\int_{0.4}^{0.76} d\lambda \cdot t(\lambda) \cdot I_{AM1.5}(\lambda)}{\int_{0.4}^{0.76} d\lambda \cdot I_{AM1.5}(\lambda)} \tag{3}$$

where $I_{AM1.5}(\lambda)$ represents the solar illumination of the AM 1.5 spectrum and $I_{BB}(T, \lambda)$ is the spectral radiance of a blackbody at temperature $T$. $t(\lambda)$ and $\varepsilon(\lambda)$ are the transmittance and emittance, respectively, of the film in the corresponding wavelength range.

## Data availability
The authors declare that all data supporting the findings of this study are available within the article and the Supplementary Information. All

other data are available from the corresponding authors upon request. Source data are provided with this paper.

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

## Acknowledgements

The authors thank the Natural Science Foundation of Henan (242300421010), National Key R&D Program of China (2019YFA0706802, 2021YFB3200302, and 2021YFB3200304), National Natural Science Foundation of China (52125205, U20A20166, and 52192614), Natural Science Foundation of Beijing Municipality (Z180011 and 2222088), Shenzhen Science and Technology Program (Grant No. KQTD20170810105439418) and the Fundamental Research Funds for the Central Universities for financial support. X.L. would like to thank Dr. Fenghua Shi, Dr. Chengfeng Du, Dr. Kun Liang for their guidance.

## Author contributions

X.L. and C.P. conceived and designed the experiments. W.Z. and X.Z. conducted the experiments and characterizations and analyzed the data with the help of Y.P., C.W., C.S., X.L., C.L., and C.P., Z.Z. performed the simulations with the help of B.H., C.P., and C.L. The first manuscript was drafted by X.Z., W.Z., X.L., and C.P. with input from all the other authors.

## Competing interests

The authors declare no competing interests.
