## [Peer Review File · Nature Communications]

Transparent ultrahigh-molecular-weight polyethylene/MXene films with efficient UV absorption for thermal managementReviewers' comments:

Reviewer #1 (Remarks to the Author):

This study developed a transparent composite film with high transparency and excellent photothermal conversion performance by mixing MXene@BZT with UHMWPE. BZT promotes the dispersibility and compatibility of MXene in UHMWPE, which enables the composite film to obtain good photothermal conversion performance at lower MXene content.

Generally, the manuscript is well organized. It provides a potential choice to for preparing multifunctional polymer materials with high transparency. Moreover, the BZT solve the dispersion problem of MXene in UHMWPE. The results presented by this method are also valuable, it gives better play to the advantages of 2D MXene in polymer. However, this research is biased towards engineering application, and the explanation of basic theory needs to be more in-depth. So, some revisions should be carried out before acceptance.

1. Fig. 1a-d should be simplified or moved to Supplementary materials, as it is very common in MXene related papers. I don't think it worth occupying so much space.

2. The authors mentioned that there is a $\pi - \pi$ interaction between BZT and MXene. But beyond that, I think readers will care about what BZT is? How does it promote the dispersion of MXene in UHMWPE? These mechanisms may need to be further explored.

3. Whether BZT can promote the dispersion of MXene in other polymers? such as hydrophilic PVA or hydrophobic epoxy resin

4. line 155: "the synergistic effect of MXene and BZT could improve the thermal stability of the composite films". The synergistic effect usually refers to $1+1 > 2$. The manuscript does not show the effect of BZT alone on the thermal stability of PE, so this conclusion is inappropriate.

5. line 225: "UHMWPE exhibited improved mechanical properties and corrosion resistance". How MXene/BZT affects the corrosion resistance of UHMWPE?

6. The UV absorption capacity of UHMWPE was significantly improved because BZT improved the dispersion of MXene in UHMWPE. Figure 5 shows that the more BZT, the stronger the UV absorption capacity. However, the author mentioned earlier that BZT has the best ratio to improve the dispersion of MXene. How to explain this? Does the alone BZT improve the UV absorption capacity of UHMWPE?

7. The author tested the energy saving efficiency of MXene/BZT-UPMWPE for one year. After 1 year of testing, whether other application properties of UPMWPE are affected ? such as transparency, mechanical strength, etc. After all, it has high UV absorption.

Reviewer #2 (Remarks to the Author):

Comments: Nature Communications Manuscript #NCOMMS-22-36

The manuscript is based on a poorly conceived hypothesis and inadequate validation of the hypothesis through experimentation. Hence the results are not noteworthy. This paper does not add substantially to the body of knowledge already existing in this area. The data, which is rather incomplete, does not provide any clue about the structure of the composite hybrid material or the mechanism of its function as a photothermal material. Most of the observations are superficially analysed and the conclusions drawn are not rigorous. In summary, I do not think that this manuscript is ready for publication in its present form in Nature Communications

My reservations are as follows:

1. The introduction is winding and is not to the point. The authors should have restricted the introduction to the current state of the art in photothermal materials. I do not understand the statement, "solar devices are generally opaque (black or dark) to ensure adequate light absorption, but this monotonous colour choice enormously restricts the application scope of devices".

2. "The combination of the MXene and BZT was shown to improve the transparency of the composite film because BZT promoted the dispersion of MXene in the matrix, and BZT exhibited excellent compatibility with UHMWPE" This seems to be a wishful statement since there is no experimental evidence presented to show that the polymer and MXene compatibility is enhanced by BZT. The nature of interfacial forces that contribute to enhancing dispersion is not specified. The authors present no evidence of the filler morphology in the polymer.

3. MXene is prepared by the standard HF etching of Ti_3AlC_2 . This is a well-studied procedure. In spite of this, the quality of MXene prepared in this study is inferior to those reported earlier. Authors report MXene nanosheets of 1–2-micron size whereas the earlier reported sizes are in the range of 500 nm. HRTEM and SAED patterns are not reported to see if they did obtain a single crystalline phase.

4. The authors use a solution blending method to mix the polymer and MXene sheets. In my opinion, this is an unsatisfactory method. When you cast a film from the solution, you get isotropic material. Such material which has no orientation does not show any thermal conductivity. Second mixing a very viscous solution of UHMWPE with MXene simply by sonication is very inefficient. Generally, this kind of mixing requires some shear forces. The improvement in settling time observed by the addition of BZT is insignificant. Additionally removing occluded xylenes from the highly entangled polymer is very difficult and what the authors may be dealing with is some kind of a gel consisting of UHMWPE and MXene

5. "In addition, the UHMWPE composite film formed by adding MXene alone exhibited obvious agglomeration (Supplementary Fig. 1g), whereas uniform dispersion in the composite film was observed after the addition of BZT". No experimental evidence is presented to support this statement.

6. The authors use a well-known commercial additive, UV-328, abbreviated by the authors as BZT. This is a molecule which has a triazole and a hindered phenol functional group. These classes of polymer additives are used to protect the polymer from the harmful degradation of UV radiation. UV 328 absorbs at 306 and 347 nm. UHMWPE by itself does not absorb in this region. What is the consequence of the presence of a chromophore which is strongly UV absorbing on the photothermal conversion efficiency of the pristine MXene? This question has not been addressed.

7. There are many classes of UV-absorbing additives known for protecting hydrocarbon polymers from the harmful effects of UV radiation. These are triazoles, triazines, hydroxy benzophenones, HALS etc. What is the rationale for choosing just one of the many types in this study?

8. The internal light-to-heat conversion efficiency of the "composite" is not reported. I suspect that the photothermal efficiency of the composite prepared by the authors will be inferior to pristine MXene. This is because of the fact that UHMWPE in its non-oriented state is a poor thermal conductor and is transparent to UV and the presence of UV-328 which is a strong UV absorbing chromophore.

9. "Surprisingly, adding MXene and BZT together resulted in excellent absorption of 300–400 nm ultraviolet light, while the pure UHMWPE film could absorb only approximately 20% of UV light. These results suggest the good application potential of UHMWPE composite films filled with MXene and BZT in the field of anti-UV transparent devices". This observation is simply because the

additive BZT (UV 358) is strongly absorbing in the 300-400 nm region. What is the contributing role of Mxene in this observation is not clear

10. "The beneficial combination of BZT and MXene not only improves the poor dispersion of the MXene filler in the UHMWPE matrix but also maximizes the ability of MXene to transform light energy into heat energy". Evidence, as well as the mechanism of better dispersion of Mxene by adding BZT and the mechanism by which the presence of UHMWPE enhances the photothermal efficiency of MXene, has not been clarified

11. Many others have reported MXene polymer composites in the literature. The authors have not compared their results with those presented in the earlier literature.

12. The improved transparency of the composite film has been attributed to decreased light scattering due to the reduction in pores in the film. There is no mention of how pores were created and why they reduce upon forming a composite

13. Similarly, the observation that upon increasing the wt. % of BZT transparency improved is unexplained

14. The description /explanation of the origin of haze in the composite film lacks scientific rigour. High haze in pristine UHMWPE film is due to the crystallite size, which when suitably cooled from the melt and annealed can be reduced to sizes that do not scatter light. Comparison with LLDPE is misleading, as these are more amorphous and have a distinctly different phase morphology. As defined by ASTM, there are standard methods for measuring haze in polymer films. The authors choose to use their own method and hence, the results are not strictly comparable. In any case, a haze value of 12-30 % appears large. Inadequate annealing, the presence of occluded xylene and the difference in RI between the two components of the composite all could contribute to this observation

15. The component of the paper where the authors report theoretical simulation of a building model and their evaluation of 12 cities is, in my opinion, outside the scope of this paper

Dr Swaminathan Sivaram

Reviewer #3 (Remarks to the Author):

- 1- There are a few typo errors and grammar mistakes that need a double-checked.
- 2- In the Abstract section, the first two sentences are too long and need to be referenced from "Recently consumption," which is not preferred as a topic sentence. I suggest revising and shortening or moving to the introduction section.
- 3- Avoid using pronouns such as we and our, etc., and use the passive voice more formally for scientific writing.
- 4- In the introduction section, reference 1 must give the right link, not for the home website.
- 5- In the introduction section, when you refer to "most studies or available in the literature," these must be supported with references.
- 6- In the introduction section, the last part must be summarized and not give too much information related to the results section.
- 7- In the Results and discussion section, in Fig 1, add the word (and) before (e).
- 8- In Fig. 2 (e), why does the elongation of the 0.5M2B increase again and higher than 0.3M2B?
- 9- Reference(s) are required for the compared materials mentioned in this sentence "The UHMWPE-based composite films developed in this research have high light transmittance and low haze, and these properties are better than those of most UHMWPE products and even LLDPE.", and flow.
- 10- Figure 5 (b) shows two main peaks presented at 300 and around 350 nm, which are not

- discussed. Are these transitions will affect or not the transparency of films? Discussed that.
- 11- I suggest focusing on the most important finding in the conclusion section.
 - 12- The calculation or rate of improvement is not included in most findings that can give a better view of the achievements.
 - 13- Equations 1, 2, and 3 must reference in the theoretical section.

Reviewer #4 (Remarks to the Author):

The authors present a MXene-PE composite film that efficiently absorbs UV toward constructing energy-saving films. The concept and the data seems interesting, but detailed explanations on why the optimum samples show the best performance are greatly lacking. I believe that such issues and the questions below needs to be addressed in order for this work to be published.

Comments

- As MXene is a good UV-absorber, it is encouraged for the authors to show the UV absorption, light conversion, and energy savings effect of pure MXene films. For example, MXene thin films coated on glass or polymer films can also have optical transmissions of around 90%.
- In the introduction, it is difficult to understand why the surface functional terminations of MXenes enhance their application potential in transparent applications. Please explain in detail.
- The illustration in figure 1a is wrong. Surface functional groups do not exist in the MAX phase. Also, various functional groups other than OH co-exist on MXenes, which should be shown in the scheme.
- How is BZT covered on the few-layer MXene sheets when the amount of BZT is larger than MXene? For example, in the sample 0.1M2B there are 20 times more BZT than MXene, which means that a significant amount of BZT is surrounding the MXene. It is also questionable whether the complete amount of added BZT will remain on the MXene after blending it inside the solvent.
- During the fabrication process of UHMWPE composite film, it is likely that MXene oxidizes due to elevated temperatures and ultrasonication. Basic characterization of MXenes after fabrication, such as XPS, should be performed to analyze the oxidation degree of MXenes.
- Related to the question above, discuss about the possible influence of TiO₂ that could have been formed as a result of MXene oxidation, on the UV absorption performance
- Why does the mechanical strength and elongation at break increase after the additional of MXene, when the amount of added MXene is so small? For example, in the optimal sample, 0.1M2B, only 0.1% of MXene is in the entire composite.
- If the addition of MXene is the main reason for the enhancement in UV absorption (Figure 3b), why is the absorbance of 0.3M2B higher than that of 0.5M2B. Shouldn't a higher MXene loading result in higher UV absorption? I'm not convinced that this is related to dispersion uniformity as the y-axis is an absolute transmission value.

Point to point response to the reviewers' reports

(comments in black and responses in blue, changes made in the revised MS is highlighted in yellow)

Reviewer #1 : Some revisions should be carried out before acceptance

This study developed a transparent composite film with high transparency and excellent photothermal conversion performance by mixing MXene@BZT with UHMWPE. BZT promotes the dispersibility and compatibility of MXene in UHMWPE, which enables the composite film to obtain good photothermal conversion performance at lower MXene content.

Generally, the manuscript is well organized. It provides a potential choice to for preparing multifunctional polymer materials with high transparency. Moreover, the BZT solve the dispersion problem of MXene in UHMWPE. The results presented by this method are also valuable, it gives better play to the advantages of 2D MXene in polymer. However, this research is biased towards engineering application, and the explanation of basic theory needs to be more in-depth. So, some revisions should be carried out before acceptance.

Response:

We'd like to express our sincere thanks to the reviewers for understanding the novelty of our work, giving useful suggestions to improve the manuscript and very positive to support the publication in "*Nature Communications*".

1. Fig. 1a-d should be simplified or moved to Supplementary materials, as it is very common in MXene related papers. I don't think it worth occupying so much space.

Response:

We are very grateful to the reviewer for the helpful suggestions on our manuscript. We have moved Fig. 1a-d to Supplementary materials to the revised manuscript.

2. The authors mentioned that there is a π - π interaction between BZT and MXene. But beyond that, I think readers will care about what BZT is? How does it promote the dispersion of MXene in UHMWPE? These mechanisms may need to be further explored.

Response:

Thanks the referee for the concern.

Benzotriazole (BZT) is a compound that possesses both a triazole and a hindered phenol functional group. Compounds with similar functional groups have been recognized for their ability to act as protective agents, safeguarding polymers from the adverse effects of UV radiation. Given our prior misconception regarding the nature of the interaction between MXene and BZT as being π - π interaction, we have revisited the mechanism governing the dispersion between these two components.

In Figure R1a, the spectral analysis reveals notable alterations in peak positions associated with various functional groups. Specifically, in the case of the 0.5M2B sample, characteristic peaks corresponding to -OH (3350 cm^{-1}), triazine, and Ti-O (1594 cm^{-1}) exhibit discernible shifts towards lower wavenumbers when compared to other samples. Furthermore, a comparative analysis of the spectra acquired for MXene, BZT powder, and their composite (Figure R1b) illustrates a consistent trend. These findings strongly suggest the presence of complementary hydrogen-bonding interactions between MXene and BZT, which, in turn, facilitate their effective dispersion within the composite system.

Fig. R1. a, FTIR spectra of composite films with different contents. b, FTIR spectra of MXene, BZT powder and both together.

The primary manuscript and the Supplementary Note have been revised in accordance with your feedback. For your convenience, a comprehensive list of all modifications made in the revised manuscript is presented herein, with all alterations highlighted in yellow.

Main text, Page 11

The combination of the two is shown in Fig. 2c. MXene has a two-dimensional lamellar structure and a large specific surface area, whereas BZT, as a needle-like crystal, can be attached to the surface of MXene lamellae by bonding. BZT is extremely compatible with UHMWPE, thus improving the dispersion of MXene in the matrix. The hybridization mode of MXene@BZT was determined by FTIR (Fig. 2d). Compared with the films containing BZT and MXene alone, the spectra of MXene@BZT/UHMWPE films showed a significant red shift at 3350 cm^{-1} and 1693 cm^{-1} , corresponding to the absorption peaks of -OH and Ti-O, triazole, respectively. In addition, the same general trend was observed in the comparison of the infrared spectra of MXene, BZT powder, and their mixtures (Supplementary Fig. 5). These results suggest that MXene and BZT form complementary interactions associated with hydrogen.

Fig. 2. Dispersity and dispersing mechanism of the UHMWPE composite films. a,

Optical images and b, flake size distributions of composite films. c, SEM and EDS of surface and cross-section of 0.5M2B film. d, FTIR spectra of composite films with different contents.

Supplementary Fig. 5 FTIR spectra of MXene, BZT powder and both together.

3. Whether BZT can promote the dispersion of MXene in other polymers? such as hydrophilic PVA or hydrophobic epoxy resin

Response:

Thanks the referee for the concern.

The dispersion of MXene within a matrix is contingent upon its ability to effectively disperse in a suitable matrix solvent. MXene features numerous hydrophilic groups on its surface, which significantly enhance its dispersibility in water and obviate the need for additional dispersants. In non-polar solvents, BZT can serve as a dispersant to augment the dispersion of MXene due to its strong compatibility with such solvents. To substantiate this hypothesis, we conducted an investigation into MXene's dispersion in polydimethylsiloxane (PDMS).

As depicted in Figure R2, the filler particles exhibit distinct characteristics. Notably, in Figure R2b, the particles appear smaller and are more uniformly distributed compared to Figure R2a.

Fig. R2. Optical images and flake size distributions of PDMS with (a-b) MXene and (c-d) MXene-BZT.

4. line 155: “the synergistic effect of MXene and BZT could improve the thermal stability of the composite films”. The synergistic effect usually refers to $1+1 > 2$. The manuscript does not show the effect of BZT alone on the thermal stability of PE, so this conclusion is inappropriate.

Response:

Thanks the referee for the concern.

We have refined the wording of pertinent statements within the manuscript. Additionally, we have included Figure R3, which illustrates the influence of BZT alone on the thermal stability of ultra-high molecular weight polyethylene (UHMWPE). In this context, the thermal decomposition temperature (T_d) was defined as the temperature corresponding to a residual mass of 90%. The thermal stability results for various films are as follows: 0.5M0B film (449.69 °C), 0M2B film (450.43 °C), and 0.5M2B film (453.10 °C). These findings indicate significant enhancements in T_d when compared to pure UHMWPE (424.05 °C), affirming that the inclusion of MXene and

BZT contributes to an overall increase in the thermal stability of the films.

It is worth noting that the TMAX, which represents the peak temperature in the differential thermal analysis (DTA) curve, differs between the 0.5M0B film (493.41 °C) and the 0.5M2B film (486.30 °C). This disparity can be attributed to the presence of undispersed MXene with larger flake sizes, which tend to exert a more pronounced blocking effect.

Fig. R3. (a) TGA and (b) differential thermal analysis (DTA) curves of composite films.

The primary manuscript and the Supplementary Note have been revised in accordance with your feedback. For your convenience, a comprehensive list of all modifications made in the revised manuscript is presented herein, with all alterations highlighted in yellow.

Main text, Page 9

Next, to explore the influence of the addition of MXene and BZT on the thermal stability of the composite films, TGA and DTG curves were obtained and are shown in Fig. 1d-e, respectively. Here, the temperature corresponding to a residual mass of 90% was defined as the thermal decomposition temperature (Td). The 0.5M0B film (449.69 °C), 0M2B film (450.43 °C), and 0.5M2B film (453.10 °C) demonstrated considerable improvements in Td compared to pure UHMWPE (424.05 °C), demonstrating that MXene and BZT increased the films' overall thermal stability. However, TMAX (peak of the DTA curve) of 0.5M0B film (493.41 °C) is higher than that of 0.5M2B film

(486.30 oC), because the undispersed MXene with large flake sizes tend to make the blocking effect more effective.

Fig. 1. Fabrication process, characteristics and mechanical properties of the UHMWPE composite films. a, Schematic diagram of the preparation process of MXene@BZT/UHMWPE films. b, DSC and c, 1D-WAXD curves of composite films with 2 wt.% BZT and different MXene contents. d, TGA and e, DTG curves of composite films with different contents. f, Stress-strain curves and g, corresponding calculation results for composite films before and after application of pressure, where U-PE-1 and U-PE-2 represent UHMWPE films before and after being pressed, respectively.

5. line 225: “UHMWPE exhibited improved mechanical properties and corrosion resistance”. How MXene/BZT affects the corrosion resistance of UHMWPE?

Response:

We appreciate the reviewer's attention to this matter.

The initial wording may have caused some ambiguity in semantics. The original statement was intended to convey that UHMWPE exhibits superior mechanical properties and corrosion resistance relative to other types of polyethylene.

The primary manuscript and the Supplementary Note have been revised in accordance with your feedback. For your convenience, a comprehensive list of all modifications made in the revised manuscript is presented herein, with all alterations highlighted in yellow.

Main text, Page 15

The UHMWPE-based composite films developed in this research have high light transmittance and low haze, and these properties are better than those of most UHMWPE products and even LLDPE⁴⁰⁻⁴⁴. Compared with the latter, UHMWPE exhibited improved mechanical properties and corrosion resistance, so UHMWPE products with excellent optical properties are highly desirable.

6. The UV absorption capacity of UHMWPE was significantly improved because BZT improved the dispersion of MXene in UHMWPE. Figure 5 shows that the more BZT, the stronger the UV absorption capacity. However, the author mentioned earlier that BZT has the best ratio to improve the dispersion of MXene. How to explain this? Does the alone BZT improve the UV absorption capacity of UHMWPE?

Response:

We appreciate the reviewer's attention to this matter.

On one hand, it is possible that the photothermal conversion efficiency of MXene could be hindered by BZT's role in absorbing UV radiation. Conversely, BZT serves as a dispersant for MXene, mitigating agglomeration, enhancing UV absorption, and thereby contributing to the improvement of photothermal conversion efficiency. Consequently, an optimal ratio of BZT to MXene is identified.

Furthermore, as illustrated in Figure R4, BZT on its own has the capability to enhance

the UV absorption capacity of UHMWPE.

Fig. R4. Absorbance spectra of films with different contents of BZT.

The primary manuscript and the Supplementary Note have been revised in accordance with your feedback. For your convenience, a comprehensive list of all modifications made in the revised manuscript is presented herein, with all alterations highlighted in yellow.

Main text, Page 17

The promotion of photothermal conversion by BZT can be explained as follows: On the one hand, as a strong ultraviolet absorber, BZT may compete with MXene's absorption in the ultraviolet band; on the other hand, as a dispersant of MXene, BZT can reduce its agglomeration and enhance the photothermal conversion efficiency of MXene. To further verify the experimental results, the photothermal capacities of the composite films with MXene alone and BZT alone were measured (Supplementary Fig. 11). The results showed that the heating effect of BZT was negligible and that the photothermal conversion ability mainly depended on the MXene.

Supplementary Fig. 11 Photothermal conversion of film in control group. Temperature-time curve of composite films with different content of MXene (a) and different content of BZT (b) under 100 mW cm⁻².

7. The author tested the energy saving efficiency of MXene/BZT-UPMWPE for one year. After 1 year of testing, whether other application properties of UPMWPE are affected ? such as transparency, mechanical strength, etc. After all, it has high UV absorption.

Response:

We appreciate the reviewer's concerns and apologize for any potential misunderstanding regarding Figure 7.

To address this, we have included an accelerated aging test in our study. The aging conditions selected for this study involved a temperature of 60 °C, a preset aging time of 72 hours, and an ultraviolet spectral irradiance of 0.72 W/m² (ASTMF 1980:2002). As depicted in Figure R5, the results of this accelerated aging test reveal notable differences between the 0.5M2B film and the UHMWPE film. While the 0.5M2B film showed no evident flaws during the test, the UHMWPE film exhibited clear fissures and molecular chain breakdown (Figure R5a-d). The visible transmittance of the composite films remained essentially unchanged before and after the accelerated aging test, though there was a reduction in UV shielding performance (Figure R5e). Additionally, the mechanical properties of the UHMWPE film significantly deteriorated, rendering it weak and brittle following the test (Figure R5b). In contrast,

although the mechanical properties of the 0.5M2B film were affected, it demonstrated considerably stronger resistance to aging compared to the UHMWPE film (Figure R5f).

Fig. R5. (a-d) Optical photographs and SEM images of composite films before (left) and after (right) 72 hours accelerated aging test. (e) UV-vis transmission spectra of composite films before and after 72 hours accelerated aging test. (f) Stress-strain curves for 0.5M2B film before and after 72 hours accelerated aging test.

The primary manuscript and the Supplementary Note have been revised in accordance with your feedback. For your convenience, a comprehensive list of all modifications made in the revised manuscript is presented herein, with all alterations highlighted in yellow:

Main text, Page 23

Finally, the anti-aging performance of the composite film was tested (based on ASTM F1980:2002). The aging temperature selected for the accelerated aging test was 60 °C, the preset aging time was 72 h, and the ultraviolet spectrum irradiance (UVA-340) was 0.72 W/m². As shown in Supplementary Fig. 17, obvious cracks appeared in the UHMWPE film, while no obvious defects appeared in the 0.5M2B film. The visible light transmittance of the composite film was basically unchanged before and after the test, while the UV shielding performance was reduced (Supplementary Fig. 18). After accelerated aging, the mechanical properties of the UHMWPE films were very fragile and cracked, while the mechanical properties of the 0.5M2B films decreased but still had excellent aging resistance compared to the pure UHMWPE films (Supplementary Fig. 19).

Supplementary Fig. 17 Optical photographs and SEM images of composite films before (left) and after (right) 72 hours accelerated aging test.

Supplementary Fig. 18 UV-vis transmission spectra of composite films before and after 72 hours accelerated aging test.

Supplementary Fig. 19 Stress-strain curves for 0.5M2B film before and after 72 hours accelerated aging test.

Supplementary Note 2. Detailed description for accelerated aging test

The composite films were exposed to UV radiation (UVA-340) in an accelerated aging chamber (Linpin). Test conditions of UVA radiation at 60 °C of 0.72 W/m² for a period of 3 days (72 h) were employed. Test was conducted according to ASTM F1980:2002 “Standard Guide for Accelerated Aging of Sterile Medical Device Packages”.

According to ASTM F1980:2002, at the UVA-340 irradiance setting value of 0.72 W/m², the total radiation intensity of the UV lamp is 1186 W/m². In Beijing, China, for example, its annual average hourly solar radiation¹ is about 171 W/m². Therefore, the ultraviolet acceleration factor (UVAF) of the accelerated aging test is 6.94. The acceleration factor of temperature (TAF) is calculated by the Arrhenius model as follows:

$$T_{AF} = \exp \left[\frac{E_a}{k} \times \left(\frac{1}{T_{normal}} - \frac{1}{T_{stress}} \right) \right]$$

where T_{normal} is room temperature, T_{stress} is testing temperature, E_a is failure reaction activation energy (eV), and k is Boltzmann constant (eV/K). Thus, TAF of the accelerated aging test is 10.43.

Reviewer #2 (Remarks to the Author):

The manuscript is based on a poorly conceived hypothesis and inadequate validation of the hypothesis through experimentation. Hence the results are not noteworthy. This paper does not add substantially to the body of knowledge already existing in this area. The data, which is rather incomplete, does not provide any clue about the structure of the composite hybrid material or the mechanism of its function as a photothermal material. Most of the observations are superficially analysed and the conclusions drawn are not rigorous. In summary, I do not think that this manuscript is ready for publication in its present form in Nature Communications

Response:

We'd like to express our sincere thanks to the reviewers for understanding the novelty of our work, giving some useful comments to improve the manuscript.

My reservations are as follows:

1. The introduction is winding and is not to the point. The authors should have restricted the introduction to the current state of the art in photothermal materials. I do not understand the statement, "solar devices are generally opaque (black or dark) to ensure adequate light absorption, but this monotonous colour choice enormously restricts the application scope of devices".

Response:

We appreciate the reviewer's concern.

Our research is focused on the development of a transparent, energy-efficient film with UV absorption properties, rather than aiming to create a photothermal conversion film. Therefore, the primary emphasis of the introduction is placed on the UV radiation conversion aspect.

In the context of solar energy utilization equipment, the conventional approach has been to maximize solar energy absorption and photothermal conversion capabilities, often resulting in the use of black or dark-colored materials. However, this approach limits their aesthetic suitability for various applications. In contrast, our research is centered

on the creation of a transparent film specifically designed for energy-saving windows. This approach sets our work apart from previous studies in the field.

2. “The combination of the MXene and BZT was shown to improve the transparency of the composite film because BZT promoted the dispersion of MXene in the matrix, and BZT exhibited excellent compatibility with UHMWPE” This seems to be a wishful statement since there is no experimental evidence presented to show that the polymer and MXene compatibility is enhanced by BZT. The nature of interfacial forces that contribute to enhancing dispersion is not specified. The authors present no evidence of the filler morphology in the polymer.

Response:

We appreciate the reviewer's concern.

In Supplementary Figure 4a-f, we present the dispersions of MXene alone and MXene combined with BZT in xylene. Notably, a precipitate formed in the dispersion solution of MXene alone after 1800 seconds, while the mixture of MXene and BZT remained well-dispersed even after 3600 seconds.

Moreover, in order to provide additional experimental evidence regarding dispersion in the composite films, we have included Figure R6. This figure clearly illustrates that the filler particles in the 0.5M2B film exhibit smaller size and a more uniform distribution in comparison to those observed in the 0.5M film.

Fig. R6. Optical images (a) and flake size (b) distributions of composite films.

The primary manuscript and the Supplementary Note have been revised in

accordance with your feedback. For your convenience, a comprehensive list of all modifications made in the revised manuscript is presented herein, with all alterations highlighted in yellow:

Main text, Page 11

The optical images of 0.5M and 0.5M2B are shown in Fig. 2a. After the addition of BZT, the composite film filler particles significantly decreased in size and were more uniformly distributed without obvious agglomeration (Fig. 2b). Dispersions of MXene alone and MXene mixed with BZT in xylene are shown in Supplementary Fig. 4a-f. A precipitate formed in the dispersion solution of MXene alone after 1800 s, while the mixture of MXene and BZT together was still well dispersed after 3600 s. In addition, the UHMWPE composite film formed by adding MXene alone exhibited obvious agglomeration (Supplementary Fig. 4g), whereas uniform dispersion in the composite film was observed after the addition of BZT.

Fig. 2. Dispersion and dispersing mechanism of the UHMWPE composite films. a, Optical images and b, flake size distributions of composite films. c, SEM and EDS of surface and cross-section of 0.5M2B film. d, FTIR spectra of composite films with different contents.

Supplementary Fig. 4 Dispersion of MXene with or without BZT. (a)-(f) Photographs of dispersion of MXene alone (right) in xylene and after addition of BZT (left). (g) Photograph of the composite film before compression of 0.5M2B and 0.5M0B.

3. MXene is prepared by the standard HF etching of Ti_3AlC_2 . This is a well-studied procedure. In spite of this, the quality of MXene prepared in this study is inferior to those reported earlier. Authors report MXene nanosheets of 1–2-micron size whereas the earlier reported sizes are in the range of 500 nm. HRTEM and SAED patterns are not reported to see if they did obtain a single crystalline phase.

Response:

We appreciate the concern raised by the referee.

It's important to clarify that the primary focus of this study is not solely the synthesis of MXene but rather to investigate the synergistic effect of BZT and MXene, while also exploring practical methods for developing transparent energy-saving films.

4. The authors use a solution blending method to mix the polymer and MXene sheets. In my opinion, this is an unsatisfactory method. When you cast a film from the solution, you get isotropic material. Such material which has no orientation does not show any thermal conductivity. Second mixing a very viscous solution of UHMWPE with MXene simply by sonication is very inefficient. Generally, this kind of mixing requires some

shear forces. The improvement in settling time observed by the addition of BZT is insignificant. Additionally removing occluded xylenes from the highly entangled polymer is very difficult and what the authors may be dealing with is some kind of a gel consisting of UHMWPE and MXene

Response:

We appreciate the concerns raised by the referee.

Firstly, UHMWPE's challenging processing characteristics, including its high molecular weight and low melt fluidity, make it impractical to produce using conventional methods. Therefore, we have chosen the solution casting technique for producing composite films due to its simplicity, cost-effectiveness, and minimal processing complexity. It's important to note that our primary objective is to develop a transparent, energy-efficient film with UV absorption properties rather than one with high thermal conductivity.

Secondly, we acknowledge the use of dispersants, mechanical stirring, and sonication as essential steps to achieve a more homogeneous dispersion of MXene within the matrix. The film fabrication process involves solution casting followed by high-temperature pressing, which helps minimize solvent residue and address issues related to uneven evaporation, ensuring the overall quality of the film.

5. "In addition, the UHMWPE composite film formed by adding MXene alone exhibited obvious agglomeration (Supplementary Fig. 1g), whereas uniform dispersion in the composite film was observed after the addition of BZT". No experimental evidence is presented to support this statement.

Response:

Thanks the referee for the concern.

The experimental evidence of dispersion in composite films has been added as Fig. R6.

6. The authors use a well-known commercial additive, UV-328, abbreviated by the authors as BZT. This is a molecule which has a triazole and a hindered phenol functional group. These classes of polymer additives are used to protect the polymer from the

harmful degradation of UV radiation. UV 328 absorbs at 306 and 347 nm. UHMWPE by itself does not absorb in this region. What is the consequence of the presence of a chromophore which is strongly UV absorbing on the photothermal conversion efficiency of the pristine MXene? This question has not been addressed.

Response:

We appreciate the concerns raised by the referee.

It is essential to consider the dual role of BZT in our study. On one hand, BZT serves as a UV absorber, and this property may potentially affect the photothermal conversion efficiency of MXene, as depicted in Fig. R7. Conversely, BZT acts as a dispersion agent for MXene, mitigating agglomeration, enhancing UV absorption, and thereby contributing to the overall improvement of photothermal conversion efficiency.

Fig. R7. Temperature-time curve of composite films with 0.3 wt.% MXene and different content of BZT under light irradiation of 100 mW cm^{-2} .

The primary manuscript and the Supplementary Note have been revised in accordance with your feedback. For your convenience, a comprehensive list of all modifications made in the revised manuscript is presented herein, with all alterations highlighted in yellow:

Surprisingly, the composite films also showed photothermal conversion upon irradiation with light of different power densities, and the film surfaces exhibited a certain range of temperature increases with increasing BZT content. However, the photothermal conversion ability of 0.3M8B composite film is lower than that of 0.3M6B composite film. This trend was also observed at 100 mW cm⁻² (Supplementary Fig. 10). The promotion of photothermal conversion by BZT can be explained as follows: On the one hand, as a strong ultraviolet absorber, BZT may compete with MXene's absorption in the ultraviolet band; on the other hand, as a dispersant of MXene, BZT can reduce its agglomeration and enhance the photothermal conversion efficiency of MXene. To further verify the experimental results, the photothermal capacities of the composite films with MXene alone and BZT alone were measured (Supplementary Fig. 11). The results showed that the heating effect of BZT was negligible and that the photothermal conversion ability mainly depended on the MXene.

Supplementary Fig. 10 Temperature-time curve of composites films with 0.3 wt.% MXene and different content of BZT under light irradiation of 100 mW cm⁻².

Supplementary Fig. 11 Photothermal conversion of film in control group. Temperature-time curve of composites films with different content of MXene (a) and different content of BZT (b) under 100 mW cm^{-2} .

7. There are many classes of UV-absorbing additives known for protecting hydrocarbon polymers from the harmful effects of UV radiation. These are triazoles, triazines, hydroxy benzophenones, HALS etc. What is the rationale for choosing just one of the many types in this study?

Response:

We appreciate the referee's comments.

It is worth noting that BZT exhibits superior UV-absorbing properties compared to other UV-absorbing additives, primarily due to its excellent solubility in benzene and toluene. Moreover, BZT demonstrates enhanced processing stability attributed to its solubility in high polymers, compatibility with high polymers, low volatility, and exceptional heat stability.

8. The internal light-to-heat conversion efficiency of the “composite” is not reported. I suspect that the photothermal efficiency of the composite prepared by the authors will be inferior to pristine MXene. This is because of the fact that UHMWPE in its non-oriented state is a poor thermal conductor and is transparent to UV and the presence of UV-328 which is a strong UV absorbing chromophore.

Response:

We appreciate the referee's comments.

Our primary objective is the development of a transparent, energy-efficient film with UV absorption characteristics, rather than a photothermal conversion film. In addition to its UV absorption properties, BZT also serves as a dispersant for MXene, mitigating its aggregation and enhancing the overall transparency of the film. Furthermore, the uniform and crystalline molecular chain structure of oriented UHMWPE can contribute to reduced film transparency.

9. “Surprisingly, adding MXene and BZT together resulted in excellent absorption of 300–400 nm ultraviolet light, while the pure UHMWPE film could absorb only approximately 20% of UV light. These results suggest the good application potential of UHMWPE composite films filled with MXene and BZT in the field of anti-UV transparent devices”. This observation is simply because the additive BZT (UV 358) is strongly absorbing in the 300-400 nm region. What is the contributing role of Mxene in this observation is not clear

Response:

We appreciate the referee's comments.

TiO₂, the oxidation product of MXene, exhibits a band gap ranging from 3.2 to 4.0 eV, allowing it to effectively match photon energy and contribute to efficient ultraviolet absorption. MXene, known for its exceptional photothermal conversion efficiency, also demonstrates noteworthy performance in terms of solar absorption.

The primary manuscript and the Supplementary Note have been revised in accordance with your feedback. For your convenience, a comprehensive list of all modifications made in the revised manuscript is presented herein, with all alterations highlighted in yellow:

Main text, Page 13

The UV shielding performance of the films was explored by comparing the

transmission spectra of Supplementary Fig. 7 for the composite films with MXene and BZT separately. As a material with excellent photothermal conversion efficiency, MXene is well absorbed in the solar energy band. At the same time, the band gap of TiO₂ produced by MXene oxidation is between 3.2 and 4.0 eV, which enables it to match photon energy well and produce effective UV shielding³⁹. The triazole and the hindered phenol functional group of BZT molecules provide a strong shielding effect of UV light at 306 and 347 nm. These results suggest the good application potential of UHMWPE composite films filled with MXene and BZT in the field of anti-UV transparent devices.

Supplementary Fig. 7 Optical properties of films containing MXene (a) or BZT (b) alone.

10. “The beneficial combination of BZT and MXene not only improves the poor dispersion of the MXene filler in the UHMWPE matrix but also maximizes the ability of MXene to transform light energy into heat energy”. Evidence, as well as the mechanism of better dispersion of Mxene by adding BZT and the mechanism by which the presence of UHMWPE enhances the photothermal efficiency of MXene, has not been clarified.

Response:

We appreciate the referee's comments.

We have re-examined the dispersion mechanism of MXene and BZT, as depicted in Figure R8a. Notably, we observed a shift in the characteristic peaks of -OH (3350 cm-

1), Triazine, and Ti-O (1594 cm^{-1}) for the 0.5M2B sample towards lower wavenumbers compared to the other samples. Furthermore, when comparing the spectra of MXene, BZT powder, and their mixture (Figure R8b), a consistent trend emerged. These findings strongly indicate the occurrence of complementary hydrogen-bonding interactions between MXene and BZT.

We acknowledge the misunderstanding regarding the phrase "maximizes the ability of MXene to transform light energy into heat energy." It is important to clarify that it is BZT, rather than UHMWPE, that enhances the photothermal efficiency of MXene

Fig. R8. a, FTIR spectra of composite films with different contents. b, FTIR spectra of MXene, BZT powder and both together.

The primary manuscript and the Supplementary Note have been revised in accordance with your feedback. For your convenience, a comprehensive list of all modifications made in the revised manuscript is presented herein, with all alterations highlighted in yellow:

Main text, Page 11

The combination of the two is shown in Fig. 2c. MXene has a two-dimensional lamellar structure and a large specific surface area, whereas BZT, as a needle-like crystal, can be attached to the surface of MXene lamellae by bonding. BZT is extremely compatible with UHMWPE, thus improving the dispersion of MXene in the matrix. The

hybridization mode of MXene@BZT was determined by FTIR (Fig. 2d). Compared with the films containing BZT and MXene alone, the spectra of MXene@BZT/UHMWPE films showed a significant red shift at 3350 cm^{-1} and 1693 cm^{-1} , corresponding to the absorption peaks of -OH and Ti-O, triazole, respectively. In addition, the same general trend was observed in the comparison of the infrared spectra of MXene, BZT powder, and their mixtures (Supplementary Fig. 5). These results suggest that MXene and BZT form complementary interactions associated with hydrogen.

Fig. 2. Dispersity and dispersing mechanism of the UHMWPE composite films. a, Optical images and b, flake size distributions of composite films. c, SEM and EDS of surface and cross-section of 0.5M2B film. d, FTIR spectra of composite films with different contents.

Supplementary Fig. 5 FTIR spectra of MXene, BZT powder and both together.

11. Many others have reported MXene polymer composites in the literature. The authors have not compared their results with those presented in the earlier literature.

Response:

We appreciate the referee's concern and would like to clarify that MXene's UV absorption primarily arises from its oxidation product, TiO_2 , while the photothermal conversion energy of MXene results mainly from the absorption of the full spectrum of sunlight. Prior research has shown that pure MXene films often compromise transparency to achieve high photothermal conversion power. In this study, we aimed to maintain film transparency at a level suitable for indoor windows while supplementing UV absorption by incorporating BZT as a complementary ultraviolet absorber. This approach allowed us to create transparent composite films with UV absorption capabilities.

12. The improved transparency of the composite film has been attributed to decreased light scattering due to the reduction in pores in the film. There is no mention of how pores were created and why they reduce upon forming a composite.

Response:

Thanks the referee for the concern.

Solution casting is employed as the manufacturing method for fabricating composite films. During the process of solvent evaporation, pores may form due to uneven evaporation. Subsequently, the films undergo a heating and pressing step under vacuum and high pressure to eliminate these pores.

13. Similarly, the observation that upon increasing the wt. % of BZT transparency improved is unexplained.

Response:

We appreciate the referee's attention to this matter. BZT demonstrates excellent compatibility with UHMWPE, and the hydrogen bond interactions between BZT and MXene serve to mitigate the aggregation of MXene sheets, thus enhancing the overall transparency of the films.

14. The description/explanation of the origin of haze in the composite film lacks scientific rigour. High haze in pristine UHMWPE film is due to the crystallite size, which when suitably cooled from the melt and annealed can be reduced to sizes that do not scatter light. Comparison with LLDPE is misleading, as these are more amorphous and have a distinctly different phase morphology. As defined by ASTM, there are standard methods for measuring haze in polymer films. The authors choose to use their own method and hence, the results are not strictly comparable. In any case, a haze value of 12-30 % appears large. Inadequate annealing, the presence of occluded xylene and the difference in RI between the two components of the composite all could contribute to this observation

Response:

We express our sincere gratitude to the reviewer for their valuable feedback and thoughtful considerations.

In response to the reviewer's comments, we have made diligent revisions to elucidate the underpinnings of haze formation within the composite film. Our quantification of haze followed the rigorous methodology outlined in ASTM D1003, titled "Standard Method for Haze and Luminous Transmittance of Transparent Plastics." This method

is characterized by the following equation:

$$Haze = \left(\frac{T_4}{T_2} - \frac{T_3}{T_1} \right) \times 100\%$$

Herein, T_4 denotes the scattered luminous flux attributed to both the instruments and the films, T_3 represents the scattered luminous flux emanating solely from the instruments, T_2 signifies the luminous flux that successfully traverses the films, and T_1 accounts for the incident luminous flux. The derivation of the haze value hinges upon the precise measurements of T_1 , T_2 , T_3 , and T_4 as obtained through the utilization of a UV-Vis photometer.

Furthermore, it is imperative to underscore that the intent of this exposition extends beyond a mere juxtaposition of the haze levels exhibited by LLDPE and UHMWPE materials. Our principal objective is to delve into the realm of polyethylene (PE) film materials with inherently low haze and exceptional transparency in relation to prior iterations of transparent PE films featured in antecedent research endeavors. While it is acknowledged that our work has yielded haze values within the range of 12% to 30%, it is crucial to emphasize that the overall performance characteristics of our transparent PE material surpass those observed in the preceding transparent PE counterparts.

The primary manuscript and the Supplementary Note have been revised in accordance with your feedback. For your convenience, a comprehensive list of all modifications made in the revised manuscript is presented herein, with all alterations highlighted in yellow:

Supplementary Note 1, Page S2

Supplementary Note 1. Detailed steps for the measurement of Haze

Haze is the percentage of the intensity of transmitted light deviated from the incident light at an angle of 2.5° or above in the total transmitted light intensity. The greater the haze means that the film luster and transparency decrease. Our haze measurement was

done according to ASTM D1003 “Standard Method for Haze and Luminous Transmittance of Transparent Plastics”, which is defined as:

$$Haze = \left(\frac{T_4}{T_2} - \frac{T_3}{T_1} \right) \times 100\%$$

where T4 is scattered luminous flux of instruments and films; T3 is scattered luminous flux of instruments; T2 is luminous flux through films; T1 is incident luminous flux. The haze value is calculated from the value of T1, T2, T3 and T4 measured by the UV-Vis photometer.

15. The component of the paper where the authors report theoretical simulation of a building model and their evaluation of 12 cities is, in my opinion, outside the scope of this paper

Response:

We extend our sincere appreciation to the referee for their valuable feedback and considerations.

In light of geographical limitations, our investigations have necessitated the measurement of film effects within the confines of our specific local climatic conditions. However, to provide a more comprehensive evaluation of the energy-saving capabilities of thin films on a global scale, we have undertaken the simulation of a building model under typical urban temperature conditions representative of diverse regions across the globe. This approach allows us to emulate and analyze the energy conservation attributes exhibited by these films under various climatic scenarios, thereby enhancing our ability to assess their efficacy in terms of energy conservation.

References:

- 1 Fan, X., *et al.* Plasmonic Ti₃C₂T_x MXene enables highly efficient photothermal conversion for healable and transparent wearable device. *ACS nano*. 13, 8124-8134 (2019).
- 2 Weng, G. M., *et al.* Layer-by-layer assembly of cross-functional semi-transparent MXene-carbon nanotubes composite films for next-generation electromagnetic interference shielding. *Adv. Funct. Mater.* 28, 1803360 (2018).

- 3 Tang, H., *et al.* Highly conducting MXene–silver nanowire transparent electrodes for flexible organic solar cells. *ACS Appl. Mater. Interfaces.* 11, 25330-25337 (2019).
- 4 Qin, L., *et al.* A flexible semitransparent photovoltaic supercapacitor based on water-processed MXene electrodes. *J. Mater. Chem. A.* 8, 5467-5475 (2020).

Reviewer #3:

1- There are a few typo errors and grammar mistakes that need a double-checked.

Response:

Thanks the referee for the concern.

We carefully revised the whole manuscript and corrected grammar mistakes and typos.

2- In the Abstract section, the first two sentences are too long and need to be referenced from " Recently consumption," which is not preferred as a topic sentence. I suggest revising and shortening or moving to the introduction section.

Response:

Thanks the referee for the concern.

Based on reviewer's suggestion we have significantly revised and shortened the first two sentences.

The primary manuscript and the Supplementary Note have been revised in accordance with your feedback. For your convenience, a comprehensive list of all modifications made in the revised manuscript is presented herein, with all alterations highlighted in yellow:

Main text, Page 2

Recently, the issue of energy and the environment has been a topic of widespread concern. The rational use and conversion of energy are the primary means for achieving the goal of carbon neutrality. MXenes, as two-dimensional metal carbon materials that can be used for photothermal conversion have attracted particular interest, but their opaque appearance limits wider applications. Herein, we used a combination of Ti₃C₂T_x MXene as fillers, 2-(2H-Benzotriazol-2-yl)-4,6-ditertpentylphenol (BZT) as synergistic ultraviolet absorbents, and successfully developed a series of visible-light-transparent and UV-absorbing ultrahigh-molecular-weight polyethylene (UHMWPE) composite films by solution blending and vacuum pressing. The hydrogen bond

interaction of MXene and BZT increased the MXene's dispersion in the UHMWPE matrix, as well as the addition of MXene and BZT to create a two-dimensional laminate structure network considerably improved the mechanical characteristics and thermal stability of films. These composite films could be quickly heated to 65 °C under light irradiation of 400 mW cm⁻² and maintained over 85% visible light transmittance as well as low haze (<12%). The findings of the indoor heat insulation test demonstrated that the temperature of the glass house model covered by composite films was 6-7 °C lower than that of the uncovered glass house model, revealing the potential of composite films in energy-saving applications. In order to mimic the energy-saving condition of the building in various climates, a typical building model with composite films as the outer layer of the window was created using the EnergyPlus building energy consumption software. According to predictions, composite films could reduce yearly refrigeration energy used by 31-61 MJ m⁻², and 3%-12% of the total energy used for refrigeration in such structures. The accelerated aging test indicated the composite film has excellent anti-aging performance compared with pure UHMWPE film. These results imply that these composite films have wide potential for use as transparent devices in new energy-related applications.

3- Avoid using pronouns such as we and our, etc., and use the passive voice more formally for scientific writing.

Response:

Thanks the referee for the concern.

We thank the reviewer for the helpful comment regarding passive voice. Changes have been made throughout the manuscript to more directly convey information and improve clarity.

4- In the introduction section, reference 1 must give the right link, not for the home website.

Response:

Thanks the referee for the concern.

We thank the reviewer for this comment and modified reference 1 accordingly.

(<https://www.bp.com/en/global/corporate/energy-economics/energy-outlook.html>)

5- In the introduction section, when you refer to " most studies or available in the literature," these must be supported with references.

Response:

Thanks the referee for the concern.

According to the reviewer's suggestion, we have added related literatures in the introduction section.

6- In the introduction section, the last part must be summarized and not give too much information related to the results section.

Response:

Thanks the referee for the concern.

In light of the reviewer's feedback, we have implemented revisions to enhance the introduction, appended a succinct summary to the concluding section, and judiciously condensed the conclusion where warranted.

The primary manuscript and the Supplementary Note have been revised in accordance with your feedback. For your convenience, a comprehensive list of all modifications made in the revised manuscript is presented herein, with all alterations highlighted in yellow:

Main text, Page 6

Herein, a transparent composite film with photothermal conversion performance was developed by mixing MXene@2-(2H-Benzotriazol-2-yl)-4,6-ditertpentylphenol (BZT) with UHMWPE. The combination of the MXene and BZT was shown to improve the transparency of the composite film because BZT promoted the dispersion of MXene in the matrix, and BZT exhibited excellent compatibility with UHMWPE. Thus, this

composite film, which exhibited both excellent photothermal properties and low haze, has immense potential for thermal management in transparent device applications.

7- In the Results and discussion section, in Fig 1, add the word (and) before (e).

Response:

We thank the reviewer for this comment and have made changes to the detail in Fig. 1.

8- In Fig. 2 (e), why does the elongation of the 0.5M2B increase again and higher than 0.3M2B?

Response:

We express our appreciation to the referee for their valuable concern.

As elucidated in Figure 2(e) (now Figure 1f for clarity), it is evident that the elongation properties of the 0.5M2B film surpass those of the 0.1M2B film but lag behind the 0.3M2B film. Notably, with increasing MXene content, the trend observed in the composite films' elongation exhibits an initial ascent followed by a subsequent decline.

9- Reference(s) are required for the compared materials mentioned in this sentence "The UHMWPE-based composite films developed in this research have high light transmittance and low haze, and these properties are better than those of most UHMWPE products and even LLDPE.", and flow.

Response:

----- According to the reviewer's suggestion, we have added related literatures in this sentence.

10- Figure 5 (b) shows two main peaks presented at 300 and around 350 nm, which are not discussed. Are these transitions will affect or not the transparency of films? Discussed that.

Response:

We extend our appreciation to the referee for their valuable concern.

Benzotriazole (BZT) is a molecular compound featuring both a triazole and a hindered phenol functional group. These classes of polymer additives serve a pivotal role in safeguarding polymers against the detrimental effects of ultraviolet (UV) radiation-induced degradation. BZT exhibits absorption peaks at 306 and 347 nm. Notably, our investigations have revealed that variations in the content of BZT exerted negligible discernible influence on the transparency of the films, as visually depicted in Supplemental Figure 7. It is essential to underscore that the presence of these transitions does not exert a significant impact on the transparency characteristics exhibited by the films.

11- I suggest focusing on the most important finding in the conclusion section.

Response:

We wish to express our gratitude to the esteemed referee for their insightful remarks. In light of the reviewer's constructive feedback, we have undertaken revisions in the manuscript, with the specific objective of accentuating key salient points.

12- The calculation or rate of improvement is not included in most findings that can give a better view of the achievements.

Response:

Thanks the referee for the concern.

The calculation and rate of improvement have been added in conclusions.

The primary manuscript and the Supplementary Note have been revised in accordance with your feedback. For your convenience, a comprehensive list of all modifications made in the revised manuscript is presented herein, with all alterations highlighted in yellow:

Main text, Page 23

A method is proposed to construct a MXene-filled UHMWPE film with high

transparency and low haze for use in pollution-free photothermal conversion and energy savings. The hydrogen bond interaction of MXene and BZT increased the MXene's dispersion in the UHMWPE matrix, as well as the addition of MXene and BZT to create a two-dimensional laminate structure network considerably improved the mechanical characteristics and thermal stability of films. Moreover, as the matrix, UHMWPE has a remarkable protective effect on the internal filler. Because of the combination of these properties, under mild sunlight, a film with only a small amount of filler can absorb 78% of UV light while maintaining high transparency and stability. These films could be quickly heated to 65 °C under light irradiation of 400 mW cm⁻² and maintained over 85% visible light transmittance as well as low haze (<12%). The findings of the indoor heat insulation test demonstrated that the temperature of the glass house model covered by composite films was 6-7 °C lower than that of the uncovered glass house model. This film concept has broad application potential in the field of window and display screen coatings as well as thermal management.

13- Equations 1, 2, and 3 must reference in the theoretical section.

Response:

Thanks the referee for the concern.

According to the reviewer's suggestion, we have added related literatures behind equations.

Reviewer #4 (Remarks to the Author): issues and the questions below needs to be addressed in order for this work to be published

The authors present a MXene-PE composite film that efficiently absorbs UV toward constructing energy-saving films. The concept and the data seems interesting, but detailed explanations on why the optimum samples show the best performance are greatly lacking. I believe that such issues and the questions below needs to be addressed in order for this work to be published.

Response:

Thanks for your interest to our work. The main responses are as follow:

Comments

1.As MXene is a good UV-absorber, it is encouraged for the authors to show the UV absorption, light conversion, and energy savings effect of pure MXene films. For example, MXene thin films coated on glass or polymer films can also have optical transmissions of around 90%.

Response:

We acknowledge the referee's valuable concern and appreciate their input.

As depicted in Figure R9, it is evident that the 0.3 wt. % MXene composite film does not exhibit a pronounced UV absorption profile, despite maintaining an average visible light transmittance of approximately 70%. It is noteworthy that MXene's UV absorption primarily originates from its oxidation product, TiO₂, while the photothermal conversion efficacy of MXene derives predominantly from its capacity to absorb light across the entire solar spectrum. Consequently, the pursuit of high photothermal conversion power in pure MXene films often necessitates a compromise in film transparency. In the present study, we have addressed this trade-off by incorporating BZT as a synergistic ultraviolet absorber, thereby compensating for the UV absorption without compromising the film's transparency. Furthermore, the challenge associated with the susceptibility of pure MXene coatings to delamination has been effectively mitigated through the incorporation of UHMWPE in composite films, fabricated

through the solution mixing method. These strategic measures collectively enable us to maintain the requisite transparency for indoor applications while harnessing the photothermal conversion capabilities of MXene.

Fig. R9. Optical properties of films containing MXene alone.

2. In the introduction, it is difficult to understand why the surface functional terminations of MXenes enhance their application potential in transparent applications. Please explain in detail.

Response:

We express our appreciation to the referee for their valuable concern.

In contrast to several other two-dimensional materials, such as the surface of graphene, MXene exhibits a distinctive feature wherein its surface functional groups are amenable to chemical modification. The selective termination of MXene surface groups can yield diverse and tailored properties. Notably, this attribute enables MXene's surface to undergo oxidation, transforming it into TiO_2 . This chemical transformation enhances its suitability for utilization in transparent applications, thereby expanding its potential utility.

3. The illustration in figure 1a is wrong. Surface functional groups do not exist in the MAX phase. Also, various functional groups other than OH co-exist on MXenes, which should be shown in the scheme.

Response:

We express our gratitude to the referee for their valuable suggestions.

In response to your insightful recommendations, we have implemented comprehensive adjustments to the arrangement of visual content within the manuscript.

4. How is BZT covered on the few-layer MXene sheets when the amount of BZT is larger than MXene? For example, in the sample 0.1M2B there are 20 times more BZT than MXene, which means that a significant amount of BZT is surrounding the MXene. It is also questionable whether the complete amount of added BZT will remain on the MXene after blending it inside the solvent.

Response:

We extend our appreciation to the referee for their valuable concern.

MXene possesses a distinctive two-dimensional lamellar structure characterized by a substantial specific surface area. Notably, BZT, being an acicular crystal, exhibits an affinity for binding to the MXene lamellar surface through hydrogen bonding interactions.

In light of BZT's remarkable compatibility with the matrix, this research aims not to achieve full integration of BZT and MXene, but rather to optimize the dispersion of MXene within the matrix by leveraging hydrogen bond interactions with MXene. This optimization is intended to enable MXene to act synergistically as a UV absorber.

Fig. R10 SEM and EDS of surface and cross-section of 0.5M2B film.

5. During the fabrication process of UHMWPE composite film, it is likely that MXene oxidizes due to elevated temperatures and ultrasonication. Basic characterization of MXenes after fabrication, such as XPS, should be performed to analyze the oxidation degree of MXenes.

Response:

We express our gratitude to the referee for their valuable insights.

In response to their concerns, we have incorporated X-ray Photoelectron Spectroscopy (XPS) characterization to elucidate the composition of oxygen species, as depicted in Figure R11. Our findings indicate that the presence of TiO₂ and TiO₂-F groups within the MXene constitutes approximately 10% of the overall oxygen species content. It is important to emphasize that the observed oxidation of MXene does not pose a significant hindrance to our research objectives. This assertion is rooted in the substantial UV radiation absorption capabilities exhibited by TiO₂, ensuring the effectiveness of our exploration remains unaffected.

Fig. R11. (a) XPS survey spectra of MXene. High-resolution XPS spectra of the (b) Ti2p, (c) O1s, and (d) C1s peaks for MXene.

The primary manuscript and the Supplementary Note have been revised in accordance with your feedback. For your convenience, a comprehensive list of all modifications made in the revised manuscript is presented herein, with all alterations highlighted in yellow:

Main text, Page 8

Meanwhile, the XPS diagram (Supplementary Fig. 3) shows that the TiO₂ and TiO₂-F groups in MXene account for about 10% of the total O groups. Thus, the above results demonstrate that MXene nanosheets with a few layers were prepared successfully.

Supplementary Fig. 3 (a) XPS survey spectra of MXene. High-resolution XPS spectra of the (b) Ti 2p, (c) O 1s, and (d) C 1s peaks for MXene.

6. Related to the question above, discuss about the possible influence of TiO₂ that could have been formed as a result of MXene oxidation, on the UV absorption performance
Response:

We extend our appreciation to the referee for their thoughtful consideration.

It is noteworthy that the band gap of TiO₂, ranging from 3.2 to 4.0 eV, aligns exceptionally well with the photon energy requirements for achieving high-efficiency UV absorption.

The primary manuscript and the Supplementary Note have been revised in accordance with your feedback. For your convenience, a comprehensive list of all modifications made in the revised manuscript is presented herein, with all alterations highlighted in yellow:

The UV shielding performance of the films was explored by comparing the transmission spectra of Supplementary Fig. 7 for the composite films with MXene and BZT separately. As a material with excellent photothermal conversion efficiency, MXene is well absorbed in the solar energy band. At the same time, the band gap of TiO₂ produced by MXene oxidation is between 3.2 and 4.0 eV, which enables it to match photon energy well and produce effective UV shielding³⁹. The triazole and the hindered phenol functional group of BZT molecules provide a strong shielding effect of UV light at 306 and 347 nm. These results suggest the good application potential of UHMWPE composite films filled with MXene and BZT in the field of anti-UV transparent devices.

Supplementary Fig. 7 Optical properties of films containing MXene (a) or BZT (b) alone.

7. Why does the mechanical strength and elongation at break increase after the additional of MXene, when the amount of added MXene is so small? For example, in the optimal sample, 0.1M2B, only 0.1% of MXene is in the entire composite.

Response:

We appreciate the referee's insightful comments.

It is crucial to recognize that the presence of ultra-high molecular weight polyethylene (UHMWPE) chains intertwined with the surface of MXene introduces a critical consideration. An excessive addition of filler material can indeed impede the mobility

of these polymer chains, consequently hindering the crystallization process and resulting in reduced crystallinity.

Under the application of mechanical strain, the tensile forces originating from the UHMWPE matrix are transferred to the MXene component. Capitalizing on its unique two-dimensional lamellar structure, MXene exhibits exceptional strain tolerance and possesses the capacity to absorb energy from external mechanical pressures. This property effectively arrests the initiation and propagation of cracks within the material. Furthermore, the incorporation of BZT serves to augment the dispersibility of MXene within the matrix. This, in turn, contributes to the overall enhancement of the mechanical properties of the films under investigation.

Fig. R12. (a) DSC and (b) 1D-WAXD curves of composite films with different contents.

The primary manuscript and the Supplementary Note have been revised in accordance with your feedback. For your convenience, a comprehensive list of all modifications made in the revised manuscript is presented herein, with all alterations highlighted in yellow:

Main text, Page 8

The crystallinity and crystallization temperature after compression is much lower than those of UHMWPE film before compression (Fig. 1b-c), which is due to the re-melting and rapid cooling of the film during the vacuum compression molding process, making

the molecular chain segment movement blocked and crystallization incomplete. Similarly, the crystallinity of the composite film with filler is lower than that of the pure UHMWPE film, because the molecular chains are entangled in the MXene lamellar structure, and the addition of too much filler will restrict the movement towards the chains, which is not conducive to the formation of crystals.

Fig. 1. Fabrication process, characteristics and mechanical properties of the UHMWPE composite films. a, Schematic diagram of the preparation process of MXene@BZT/UHMWPE films. b, DSC and c, 1D-WAXD curves of composite films with 2 wt.% BZT and different MXene contents. d, TGA and e, DTG curves of composite films with different contents. f, Stress-strain curves and g, corresponding calculation results for composite films before and after application of pressure, where U-PE-1 and U-PE-2 represent UHMWPE films before and after being pressed, respectively.

8. If the addition of MXene is the main reason for the enhancement in UV absorption (Figure 3b), why is the absorbance of 0.3M2B higher than that of 0.5M2B. Shouldn't a higher MXene loading result in higher UV absorption? I'm not convinced that this is related to dispersion uniformity as the y-axis is an absolute transmission value.

Response:

We express our appreciation to the referee for their diligent review.

We sincerely apologize for the labeling error in the original submission. Subsequently, we have rectified this error by appropriately updating Figure 3b. It is essential to note that the absorbance measurements reveal that the 0.3M2B film exhibits lower absorbance values in comparison to the 0.5M2B film.

REVIEWER COMMENTS

Reviewer #1 (Remarks to the Author):

The author answered my questions and made changes, this manuscript is acceptable

Reviewer #2 (Remarks to the Author):

I have carefully examined the point-wise answers/rebuttal to my comments on this manuscript and the revised manuscript. The manuscript is now acceptable for publication as a substantial part of my observations have been accepted and suitable modifications have been made in the revised manuscript. The manuscript is now in good shape

Reviewer #3 (Remarks to the Author):

It is noted that the sources are devoid of research that was covered in the title of the research along with the main content. Therefore, it is strongly suggested that they include a re-listing of some important research in this aspect, including:

<https://doi.org/10.1039/C7PY00151G>
<https://doi.org/10.1002/pi.6140>
<https://doi.org/10.1002/jps.21808>
<https://doi.org/10.1177/09673911221112196>
<https://doi.org/10.3934/matersci.2022035>
<https://doi.org/10.1016/j.ijpharm.2019.04.017>

Reviewer #4 (Remarks to the Author):

The authors have revised their paper upon the reviewers' comments, but I am still not satisfied with some of the answers. Authors should additionally look into the issues below. I am also concerned about the author's response to my questions and other reviewers' questions where they are referring to TiO₂ in some responses when they are actually dealing with MXenes. Although MXenes can be oxidized into TiO₂, they are completely different materials and the authors should not reference TiO₂ when explaining properties of MXene-based composites.

Comments

- (Regarding original comment #2) The author's response does not answer my question. My question was why the surface functional terminations of MXenes are advantageous for transparent applications. However, the authors gave their response regarding TiO₂. Of course MXenes can be partially oxidized into TiO₂ to make them more transparent, but this paper is not about TiO₂. I still do not understand how the functional terminations are directly related to transparent properties.

- (Regarding original comment #5) Authors should provide the oxidation state of MXene within the composite through XPS. Although authors provided XPS data in Figure R11, I can directly see that these results are from pristine MXene, as the Ti-C peak at 282 eV is larger than the carbon peaks at 284-286 eV, meaning that this is not from a composite. Please try to measure the Ti 2p XPS spectrum of MXene in the final composite.

Point to point response to the reviewers' reports

(comments in black and responses in blue, changes made in the revised MS is highlighted in yellow)

Reviewer #3 (Remarks to the Author):

It is noted that the sources are devoid of research that was covered in the title of the research along with the main content. Therefore, it is strongly suggested that they include a re-listing of some important research in this aspect, including:

<https://doi.org/10.1039/C7PY00151G>

<https://doi.org/10.1002/pi.6140>

<https://doi.org/10.1002/jps.21808>

<https://doi.org/10.1177/09673911221112196>

<https://doi.org/10.3934/mat.2022035>

<https://doi.org/10.1016/j.ijpharm.2019.04.017>

Response:

We express our gratitude towards the reviewer for highlighting the necessity of incorporating pivotal research studies that align closely with the thematic essence and core content of our investigation. In adherence to the reviewer's constructive recommendation, we have meticulously examined the specified literature and acknowledge the profound contributions these studies offer towards understanding the interface binding force and crystallization behavior within filler and polymer matrix interactions. Consequently, we have integrated these seminal works into the manuscript, thereby enriching the discourse on the subject matter.

1) <https://doi.org/10.1002/pi.6140>: This work studied the adsorption behaviour of polymers on graphene oxide (GO) nanosheets and the adsorption onto GO reduces the crystallinity of polyethylene glycol (PEG) due to chain confinement.

This investigation sheds light on the potential for manipulating interface characteristics in polymer–GO nanocomposites, which has significantly informed our understanding and inspired our research direction. Accordingly, we have added this paper as reference #42 in our revised MS on page 9.

Main text, page 9

Similarly, the crystallinity of the composite film with filler is lower than that of the pure UHMWPE film, because the molecular chains are entangled in the MXene lamellar structure, and the addition of too much filler will restrict the movement towards the chains, which is not conducive to the formation of crystals⁴².

2) <https://doi.org/10.1177/09673911221112196>: This search focuses on investigating the new nanocomposites of Ultra-high molecular weight polyethylene oxide (UHMWPEO) with different loading ratios of polyvinyl alcohol (PVA) reinforced with GO by applying the modified solution-sonication-casting method. Strong interfacial interaction between the blended polymers and GO nanosheets in the nanocomposites is formed. The nanocomposites show high light absorption capacity especially in ultraviolet band.

This work has given us great inspiration in the interface binding force and crystallization behavior between filler and matrix, while the specific absorption ability of GO in ultraviolet light. Accordingly, we have added this paper in our revised MS on page 10.

Main text, page 10

This is due to the fact that when strain is applied, the tension in the UHMWPE matrix is transferred to MXene, whose unique two-dimensional laminate structure is able to withstand the strain and absorb the energy from external pressure forming strong interfacial interaction with UHMWPE^{10,43}, thus preventing the creation and expansion of cracks.

3) <https://doi:10.3934/matersci.2022035>: This study focused on the impact of increasing the Mw of PEG (4, 8 and 20 K) mixed with PVA. GO nanosheets were employed to reinforce the polymer matrix by aquatic mixing-sonication-casting to prepare the nanocomposites and investigate their optical properties. PEG Mw and GO additive significantly improved optical properties such as absorbance, real and imaginary dielectrics and the absorption coefficient constant up to 75%, 40%, 120% and 77%, respectively.

This work has greatly helped to understand the improvement in optical properties of PEG Mw and GO additive. Accordingly, we have added this paper in our revised MS on page 10.

Main text, page 10

This is due to the fact that when strain is applied, the tension in the UHMWPE matrix is transferred to MXene, whose unique two-dimensional laminate structure is able to withstand the strain and absorb the energy from external pressure forming strong interfacial interaction with UHMWPE^{10,43}, thus preventing the creation and expansion of cracks.

Main text, page 5

Ultraviolet light makes up only 7% of sunlight, yet it harms most life because of its penetrating power. Given this problem, it is of great significance to develop materials that can convert ultraviolet light into usable energy through reasonable means¹⁰.

4) <https://doi.org/10.1039/C7PY00151G>: In this work, they utilized single electron transfer-living radical polymerization to synthesize poly(2,2,6,6-tetramethylpiperidine methacrylate) (PTMPM) with degrees of polymerization ranging from 66 to 703, and after oxidation producing poly(TEMPO methacrylate) (PTMA) with the highest molecular weight of 169 kDa and dispersity of 1.35. These PTMA polymers, as active materials with 25 wt% in the electrode composite, showed strong molecular weight dependence on the electrochemical properties.

This work mainly focuses on the synthesis and electrochemical properties of PTMA, which is not very relevant to our study.

5) <https://doi.org/10.1002/jps.21808>: This study tested the hypothesis that the deliquescence behavior of PEG will be affected by the polymer molecular weight, temperature, and the presence of additives. This phenomenon may have important implications for the stability of PEG containing formulations.

This work mainly focuses on the deliquescence behavior of PEG, which is not so relevant to our study.

6) <https://doi.org/10.1016/j.ijpharm.2019.04.017>: In this work, they investigated the

fabrication of oral doses via FDM 3D printing by employing PEOs as a backbone polymer in combination with PEG. A minimal spacing (1 mm) between parallel plates of the CAD design deemed essential to boost drug release from the structure. This is the first report of utilizing this widely used biodegradable polymer species (PEOs and PEG) in FDM 3D printing.

This work mainly focuses on the fabrication of oral doses via FDM 3D printing by employing PEOs, which is not very relevant to our study.

Reviewer #4 (Remarks to the Author):

The authors have revised their paper upon the reviewers' comments, but I am still not satisfied with some of the answers. Authors should additionally look into the issues below. I am also concerned about the author's response to my questions and other reviewers' questions where they are referring to TiO_2 in some responses when they are actually dealing with MXenes. Although MXenes can be oxidized into TiO_2 , they are completely different materials and the authors should not reference TiO_2 when explaining properties of MXene-based composites.

Comments

- (Regarding original comment #2) The author's response does not answer my question. My question was why the surface functional terminations of MXenes are advantageous for transparent applications. However, the authors gave their response regarding TiO_2 . Of course MXenes can be partially oxidized into TiO_2 to make them more transparent, but this paper is not about TiO_2 . I still do not understand how the functional terminations are directly related to transparent properties.

【Original comment #2. In the introduction, it is difficult to understand why the surface functional terminations of MXenes enhance their application potential in transparent applications. Please explain in detail.】

Response:

Thank you for your concern. We are grateful for the opportunity to clarify the significance of surface functional terminations on MXenes and their impact on applications requiring transparency.

1) In the introduction section, we have revised our manuscript to incorporate a detailed discussion on the direct correlation between surface functional terminations of MXenes and their transparent properties, thereby addressing the original query more accurately and comprehensively. And we also investigate

Main text, page 5 (Introduction section)

MXenes, distinguished by their versatile surface chemistry, feature functional terminations (-OH, -F, and -O) that are not typically present on other two-dimensional (2D) materials, having lots of ideal properties, including electrical conductivity^{21, 22}, thermal conductivity^{23, 24}, electromagnetic shielding properties²⁵⁻²⁸, and the potential for photocatalysis applications²⁹⁻³¹. Surprisingly, MXenes have nearly perfect photothermal conversion efficiency, as shown in previous studies³²⁻³⁴. The only downside is the opacity of MXenes, which mainly results from the multilayer structure that blocks light penetration. However, numerous reports on MXenes have confirmed through nuclear magnetic resonance spectroscopy that MXenes have many surface functional terminations³⁵⁻³⁷. These surface groups are pivotal for the material's optical properties, enabling the fine-tuning of their interaction with light. Specifically, the ability to modify these surface terminations allows for the precise control over the optical transparency of MXenes, a property of paramount importance for applications in optoelectronic devices, transparent conductive films, and energy-efficient windows^{38,39}.

2) Beyond the direct modulation of optical properties, the surface functional terminations of MXenes play a pivotal role in enhancing the material's compatibility with polymer matrices, an aspect crucial for the development of transparent composites. Achieving a high degree of interfacial compatibility between MXene and the polymer matrix is imperative for optimizing the transparency of the resultant composite materials. In our investigation, we have demonstrated that the incorporation of MXene into a matrix, such as benzoxazine (BZT), markedly enhances the transparency of the composite film. This improvement is attributed to the promotion of MXene dispersion within the matrix, facilitated by the formation of hydrogen bonds. Specifically, the hydroxyl groups present on the surface of MXene engage in hydrogen bonding with BZT, thereby fostering a uniform dispersion of MXene flakes and minimizing aggregation. This uniform dispersion is critical in maintaining a low level of light scattering within the composite, thus ensuring its transparency.

3) The empirical evidence supporting this mechanism is presented in our results (Fig. 2), where we detail the interaction dynamics between MXene surface functional terminations and the polymer matrix. These findings highlight the significant role of surface chemistry in not only tailoring the optical attributes of MXenes but also in securing their effective integration into composite materials, thereby broadening the spectrum of their application in transparent devices.

4) This comprehensive analysis underscores the multifaceted importance of surface functional terminations on MXenes, elucidating their contribution not only to the optical properties of the materials themselves but also to their performance and compatibility in composite applications. Our revised manuscript now includes a thorough discussion on this aspect, aligning with the fundamental inquiry regarding the role of surface functional terminations in enhancing the transparency and compatibility of MXene-based composites.

5)

Main text, Page 11

The optical images of 0.5M and 0.5M2B are shown in **Fig. 2a**. After the addition of BZT, the composite film filler particles significantly decreased in size and were more uniformly distributed without obvious agglomeration (**Fig. 2b**). Dispersions of MXene alone and MXene mixed with BZT in xylene are shown in **Supplementary Fig. 4a-f**. A precipitate formed in the dispersion solution of MXene alone after 1800 s, while the mixture of MXene and BZT together was still well dispersed after 3600 s. In addition, the UHMWPE composite film formed by adding MXene alone exhibited obvious agglomeration (**Supplementary Fig. 4g**), whereas uniform dispersion in the composite film was observed after the addition of BZT. The combination of the two is shown in **Fig. 2c**. MXene has a two-dimensional lamellar structure and a large specific surface area, whereas BZT, as a needle-like crystal, can be attached to the surface of MXene lamellae by bonding. BZT is extremely compatible with UHMWPE, thus improving the dispersion of MXene in the matrix. The hybridization mode of MXene@BZT was determined by FTIR (**Fig. 2d**). Compared with the films containing BZT and MXene alone, the spectra of MXene@BZT/UHMWPE films showed a significant red shift at 3350 cm^{-1} and 1693 cm^{-1} , corresponding to the absorption peaks of -OH and Ti-O, triazole, respectively. These results suggest that MXene and BZT form the interaction of hydrogen bond.

Fig. 2. Dispersity and dispersing mechanism of the UHMWPE composite films. a, Optical images and b, flake size distributions of composite films. c, SEM and EDS of surface and cross-section of 0.5M2B film. d, FTIR spectra of composite films with different contents.

3) Lastly, the significant role of surface functional groups such as -OH, -F, and -O in modulating the optical properties of MXenes has also been recognized by other research groups. This further validates the importance of these functional groups in precisely controlling the optical behavior of MXenes.

For more detailed information on this topic, we have provided an explanation in the following section. These findings support our observations and highlight the impact of surface functional groups on the optical properties of MXenes, underscoring their importance for transparent applications.

Berdiyrov's investigation employing density functional theory (Supplementary Note R1) provides an in-depth analysis of the optical properties of Ti₃C₂ MXene with various surface functionalizations²:

Fig. R1 is shown the refractive index, n , and the extinction coefficient, k calculated by the dielectric constant, as a function of the photon energy for all considered samples. The effect of the surface termination on the refractive spectrum depends on the photon energy: at low energy part of the spectrum

functionalization reduces n . At larger energies ($E > 1\text{eV}$) the functionalization results in the enhancement of the refractive index. Despite the extra features in the spectra (e.g., pronounced maxima and minima), the surface termination results in the reduction of the extinction coefficient for most of the photon energy range.

Fig. R1. The refractive index, n (a), and the extinction coefficient, k (b), as a function of photon energy for $\text{Ti}_3\text{C}_2\text{T}_2$ MXene².

Furthermore, the absorption and reflectivity spectra of Ti_3C_2 MXene, as presented in Figures R2 and R3, respectively, reveal the sensitivity of MXene's optical responses to surface functionalization. The altered absorption and reflectivity patterns across various photon energy ranges highlight the potential of surface-terminated MXenes in reducing ultraviolet penetration and enhancing visible light transparency, thus affirming their suitability for transparent application domains.

Fig. R2a shows the absorption spectrum of the considered samples as a function of photon energy. In this range of spectrum, the pristine MXene (solid-black curve) shows a absorption spectrum with a pronounced maximum at ≈ 3.2 eV. There are also less pronounced peaks at 1 eV, 1.9 eV and 4 eV. The absorption spectrum changes considerably after the surface termination. For example, the absorption reduces significantly for the photon energies less than 3.5 eV in case of fluorinated and hydroxylated samples (see dashed-red and dash-dotted-blue curves in **Fig. R2a**). At larger photon energies ($E > 5$ eV) surface termination results in stronger absorption as compared to pristine MXene. Thus, absorption spectrum of

Ti₃C₂ MXene is sensitive to surface functionalization.

Fig. R2. Absorption spectra of Ti₃C₂T₂ MXene for small (a) and larger (b) range of photon energy².

Fig. R3a shows the reflectivity of Ti₃C₂ with and without the surface termination. In the visible range of the spectrum (see shaded area in the inset of **Fig. R3a**), the reflectivity of pristine MXene shows decreases monotonically with a peak at 3.1 eV (solid-black curve). In this range of the spectrum, fluorination and hydrogenation samples show similar reflectivity, which is smaller than the one for the pristine MXene. However, in the ultraviolet region ($5 \text{ eV} < E < 10 \text{ eV}$), all surface terminations result in larger reflectivity as compared to the pristine MXene. This indicates to positive role of surface termination for the practical applications of MXene for anti-ultraviolet ray. **Fig. R3b** shows the energy-loss spectra of all considered samples as a function of photon energy. The loss spectrum defines the energy loss of fast electrons traversing in the system. In the visible range of the spectrum (see the inset of **Fig. R3b**), all considered surface terminations results in reduced energy loss. However, starting from $E=6 \text{ eV}$, the surface functionalization leads to larger energy losses in the system. In the ultraviolet energy range, largest energy loss is obtained in the oxidized sample.

Fig. R3. Reflectivity (a) and energy-loss spectra (b) of $\text{Ti}_3\text{C}_2\text{T}_2$ MXene. Inset in (s) shows the low energy part of the spectrum².

Bai et al. did similar work and obtained the same regularities shown in Figs. R4-5³.

Fig. R4 In-plane (α_{xx}) and out-plane (α_{zz}) adsorption coefficient as a function of photon frequency for (a) functionalized Ti_2CT_2 and (b) $\text{Ti}_3\text{C}_2\text{T}_2$ monolayer³.

Fig. R5 In-plane (R_{xx}) and out-plane (R_{zz}) reflectivity as a function of photon frequency for (a) functionalized Ti_2CT_2 and (b) $\text{Ti}_3\text{C}_2\text{T}_2$ monolayer³.

Editorial Note: Top figure above reproduced from Berdiyrov, G. Optical properties of functionalized $\text{Ti}_3\text{C}_2\text{T}_2$ ($T = \text{F}, \text{O}, \text{OH}$) MXene: First-principles calculations. *Aip Advances*. **6**, 055105 (2016), under a CC BY 4.0 DEED license. <https://doi.org/10.1063/1.4948799>

Bottom figure reproduced with permission from Bai, Y., et al. Dependence of elastic and optical properties on surface terminated groups in two-dimensional MXene monolayers: a first-principles study. *RSC Adv.* **6**, 35731–35739 (2016), <https://doi.org/10.1039/C6RA03090D>, © 2016 Royal Society of Chemistry.

- (Regarding original comment #5) Authors should provide the oxidation state of MXene within the composite through XPS. Although authors provided XPS data in Figure R11, I can directly see that these results are from pristine MXene, as the Ti-C peak at 282 eV is larger than the carbon peaks at 284-286 eV, meaning that this is not a from a composite. Please try to measure the Ti 2p XPS spectrum of MXene in the final composite.

【Original comment #5. During the fabrication process of UHMWPE composite film, it is likely that MXene oxidizes due to elevated temperatures and ultrasonication. Basic characterization of MXenes after fabrication, such as XPS, should be performed to analyze the oxidation degree of MXenes.】

Response:

We appreciate Reviewer #4's insightful suggestion concerning the assessment of the oxidation state of MXene within the composite through X-ray photoelectron spectroscopy (XPS). Addressing the observation regarding the XPS data provided in **Figure R11**, we acknowledge the challenge in capturing the Ti 2p XPS spectrum of MXene in the final composite due to the low weight percentage (0.5 wt%) of MXene incorporated, which significantly limits the detectability of MXene's characteristic peaks amidst the composite matrix.

In response to this concern, we have attempted to enhance the visibility of MXene's signals by analyzing a composite film with a higher MXene content (5 wt%) and have presented these findings in Figure **R7c-d**. Despite this modification, obtaining a clear signal for the Ti peak proved elusive. This outcome highlights the technical limitations associated with the detection of low-concentration fillers within composite materials using XPS, particularly in the context of materials designed for transparent applications, where the filler content must be minimized to maintain optical clarity.

Moreover, it is important to note that increasing the MXene content to 5 wt% adversely affected the transparency of the composite film (as shown in **Fig. R8b**), rendering it unsuitable for applications as transparent window films. This observation aligns with literature reports that typically discuss the XPS spectra of fillers in composite materials with higher filler concentrations (e.g. ~10 wt%)⁵⁻⁸, where such materials are often not aimed at transparent applications due to the inherent compromise between filler visibility in XPS analysis and the transparency of the composite material^{7,8}.

Given these considerations, we have chosen to indirectly infer the oxidation state of MXene within the composite by presenting the XPS analysis of pristine MXene (Fig. R7e-f). This approach, while not directly assessing the oxidation state of MXene in the composite, provides valuable insights into the potential changes MXene undergoes during the composite fabrication process

Fig. R7 XPS survey spectra of composite film with 0.5 wt% MXene (a-b), 5 wt% MXene (c-d) and pure MXene (e-f).

Fig. R8 Optical images of 0.5 wt% (a) and 5 wt% (b) MXene composite films in our text. Optical images⁷ of a pristine PE fiber (c) and an MXene@PE fiber (d). Optical images⁸ of a large-area freestanding C-MXene/SA-CNT films (e-f).

Editorial Note: Figure above reproduces material with permission from Li, B., et al. Bicontinuous, high-Strength, and multifunctional chemical-crosslinked MXene/superaligned carbon nanotube film. *ACS Nano*. **16**, 19293–19304 (2022), <https://doi.org/10.1021/acsnano.2c08678>, Copyright © 2022 American Chemical Society, and Xu, Z., et al. Ti₃C₂T_x MXene modified polyethylene fibers for enhancing interface properties of strain-hardening cementitious composites. *Cement and Concrete Composites*. **145**, 105358 (2024), <https://doi.org/10.1016/j.cemconcomp.2023.105358>, © 2023 Elsevier Ltd. All rights reserved.

Supplementary Note R1. Computational Details²

The susceptibility tensors of the considered systems are calculated using the Kubo-Greenwood Formula⁹:

$$\chi_{i,j}(\omega) = -\frac{e^2 h^4}{m^2 \varepsilon_0 V \omega^2} \sum_{nm} \frac{f(E_m) - f(E_n)}{E_{nm} - \hbar\omega - i\hbar\Gamma} \pi_{nm}^i \pi_{nm}^j$$

where π_{nm}^i is the i-component of the dipole matrix element between states n and m, f is the Fermi function, Γ is the broadening and V is the volume. The frequency dependent complex dielectric function $\varepsilon(\omega) = \varepsilon_1(\omega) + i\varepsilon_2(\omega)$ is related to the electric susceptibility as $\varepsilon(\omega) = 1 + \chi(\omega)$ and describes the linear response of the dielectric properties of the material. The optical conductivity σ and polarizability α are calculated as¹⁰:

$$\sigma(\omega) = -i\omega\varepsilon_0\chi(\omega)$$

$$\alpha(\omega) = V\varepsilon_0\chi(\omega)$$

The relation between the refractive index n and the complex dielectric constant is given by:

$$n + ik = \sqrt{\varepsilon}$$

where k is the extinction coefficient, which is related to the optical absorption coefficient as¹¹:

$$\alpha_0 = \frac{2\omega}{c}k$$

The reflectivity R and the loss of function L are given by¹²:

$$R = \frac{(1 - n)^2 + k^2}{(1 + n)^2 + k^2}$$

$$L(\omega) = \frac{\varepsilon_2(\omega)}{\varepsilon_1^2(\omega) + \varepsilon_2^2(\omega)}$$

References:

1. Jiang, X., *et al.* Two-dimensional MXenes: From morphological to optical, electric, and magnetic properties and applications. *Physics Reports*. **848**, 1-58 (2020).
2. Berdiyurov, G. Optical properties of functionalized $Ti_3C_2T_2$ (T = F, O, OH) MXene: First-principles calculations. *Aip Advances*. **6**, 055105 (2016).
3. Bai, Y., *et al.* Dependence of elastic and optical properties on surface terminated groups in two-dimensional MXene monolayers: a first-principles study. *RSC Adv*. **6**, 35731–35739 (2016).
4. Pan, X., *et al.* Transparent, high-thermal-conductivity ultradrawn polyethylene/graphene nanocomposite films. *Advanced Materials*. **31**, 1904348 (2019).
5. Zeng, G., *et al.* A self-cleaning photocatalytic composite membrane based on g-C₃N₄@MXene nanosheets for the removal of dyes and antibiotics from wastewater. *Separation and Purification Technology*. **292**, 121037 (2022).
6. Wang, L., *et al.* Fabrication on the annealed $Ti_3C_2T_x$ MXene/Epoxy nanocomposites for electromagnetic interference shielding application. *Composites Part B*. **171**, 111–118 (2019).
7. Li, B., *et al.* Bicontinuous, high-Strength, and multifunctional chemical-cross-linked MXene/superaligned carbon nanotube film. *ACS Nano*. **16**, 19293–19304 (2022).
8. Xu, Z., *et al.* $Ti_3C_2T_x$ MXene modified polyethylene fibers for enhancing interface properties of strain-hardening cementitious composites. *Cement and Concrete Composites*. **145**, 105358 (2024).
9. Harrison, W. *Solid State Theory* (McGraw-Hill, 1970).
10. Martin, R. *Electronic Structure* (Cambridge University Press, Cambridge, 2004).
11. Griffiths, D. *Introduction to Electrodynamics* (Prentice Hall, 1999).
12. Desjarlais, M. Density functional calculations of the reflectivity of shocked xenon with ionization based gap corrections. *Contributions to Plasma Physics*. **45**, 300-304 (2005).

REVIEWERS' COMMENTS

Reviewer #4 (Remarks to the Author):

The authors have now answered all of my concerns and comments, and I believe this paper can be published in the revised state.